# Detection of TurboID fusion proteins by fluorescent streptavidin outcompetes antibody signals and visualises targets not accessible to antibodies

Johanna Odenwald[1], Bernardo Gabiatti[1], Silke Braune[1], Siqi Shen[2], Martin Zoltner[2]*, Susanne Kramer[1]*

[1]Biocenter, University of Würzburg, Würzburg, Germany; [2]Department of Parasitology, Faculty of Science, Charles University in Prague, Prague, Czech Republic

**Abstract** Immunofluorescence localises proteins via fluorophore-labelled antibodies. However, some proteins evade detection due to antibody-accessibility issues or because they are naturally low abundant or antigen density is reduced by the imaging method. Here, we show that the fusion of the target protein to the biotin ligase TurboID and subsequent detection of biotinylation by fluorescent streptavidin offers an 'all in one' solution to these restrictions. For all proteins tested, the streptavidin signal was significantly stronger than an antibody signal, markedly improving the sensitivity of expansion microscopy and correlative light and electron microscopy. Importantly, proteins within phase-separated regions, such as the central channel of the nuclear pores, the nucleolus, or RNA granules, were readily detected with streptavidin, while most antibodies failed. When TurboID is used in tandem with an HA epitope tag, co-probing with streptavidin and anti-HA can map antibody-accessibility and we created such a map for the trypanosome nuclear pore. Lastly, we show that streptavidin imaging resolves dynamic, temporally, and spatially distinct sub-complexes and, in specific cases, reveals a history of dynamic protein interaction. In conclusion, streptavidin imaging has major advantages for the detection of lowly abundant or inaccessible proteins and in addition, provides information on protein interactions and biophysical environment.

*For correspondence:
martin.zoltner@natur.cuni.cz (MZ);
susanne.kramer@uni-wuerzburg.de (SK)

## eLife assessment

This **valuable** study demonstrates how proximity labeling with streptavidin can be used to boost fluorescence signals in otherwise hard-to-label regions of cells. The experimental verification of amplification of fluorescence near epitope tags in phase-separated compartments is **solid**, demonstrating enhanced signal-to-noise compared to immunofluorescence. This study will be of particular interest to those using correlative light and electron microscopy or expansion microscopy when the signal is limiting or inaccessible.

## Introduction

Visualisation of a protein within the cellular context is routinely achieved either by fusing it to a fluorescent protein, or by using fluorophore-labelled antibodies specific for a target protein or peptide tag that is genetically fused to it. Fluorescent proteins are not suitable for all imaging applications, because fluorescence is lost under fixation and permeabilisation conditions. Applications that are fully dependent on antibodies include correlative electron and light microscopy (CLEM) applications,

expansion microscopy, or protein detection in combination with in situ hybridisation for nucleic acid detection.

Sometimes, antibodies fail to detect a protein at its expected localisation, which is rarely followed up or reported. We recently observed the absence of an anti-HA antibody signal at the nuclear pores for endogenously expressed trypanosome mRNA export factor MEX67 fused to the small peptide hemagglutinin (HA) epitope tag: the anti-HA signal was entirely restricted to the nucleoplasm (*Moreira do et al., 2023*). Likewise, a polyclonal antibody raised against *T. brucei* MEX67 did not stain the nuclear pores (*Pozzi et al., 2023*). In contrast, fusions of *T. brucei* MEX67 to fluorescent proteins (either N- or C-terminally) are primarily detected at the nuclear pores with some signal extending to the nucleoplasm (*Billington et al., 2023*; *Kramer et al., 2010*). This is in agreement with the reported function of *T. brucei* MEX67 in mRNA export (*Dostalova et al., 2013*; *Schwede et al., 2009*) and with nuclear pore localisation of MEX67 orthologues in other eukaryotes (*Katahira et al., 1999*; *Köhler and Hurt, 2007*; *Mangus et al., 2003*; *Rodriguez et al., 2004*; *Stewart, 2019*; *Strässer et al., 2000*; *Terry and Wente, 2007*) where Mex67 was even termed a mobile nucleoporin (*Derrer et al., 2019*).

We next expressed *Tb*MEX67 fused to a TurboID-HA tandem tag. TurboID is a biotin ligase, widely employed for proximity labelling techniques facilitating analysis of protein interactions in vivo (*Branon et al., 2018*). To our surprise, fluorophore labelled streptavidin readily delivered a signal localizing primarily to the nuclear pores and partially to the nucleoplasm, consistent with the localisation of MEX67 fused to fluorescent proteins (*Moreira do et al., 2023*). We, therefore, concluded that the streptavidin signal shows the correct localisation of MEX67, while antibodies fail to bind MEX67 at the nuclear pore and selectively stain the nucleoplasmic fraction of MEX67. One possible reason for the absence of antibody stain could be the localisation of MEX67 to the central channel of the nuclear pore that is lined by phenylalanine-glycine nucleoporins (FG NUPs), intrinsically disordered proteins dominated by FG-repeats, that assemble to a dense meshwork and create a phase-separated environment, physically distinct from the rest of the cell (*Davis et al., 2022*; *Nag et al., 2022*). It remains unclear, why streptavidin was able to stain biotinylated proteins within these antibody inaccessible regions, but possible reasons are: (i) tetrameric streptavidin is smaller and more compact than IgGs (60 kDa versus a tandem of two IgGs, each with 150 kDa) (ii) the interaction between streptavidin and biotin is ~100 fold stronger than a typical interaction between antibody and antigen and (iii) streptavidin contains four fluorophores, in contrast to only one per secondary IgG.

Here, we set out to investigate whether streptavidin imaging offers a general solution to antibody-accessibility problems. We expressed 11 trypanosomes and one mammalian protein of different *bona fide* phase-separated regions (nucleolus, stress granules, central channel of the nuclear pores) fused to a TurboID-HA tandem tag. All proteins could be readily visualised with streptavidin, while they could not be labelled with anti-HA. Importantly, we noticed that streptavidin imaging has further major advantages to antibody labeling, even for antibody-accessible proteins: (i) the multiple biotinylation sites of the bait and adjacent proteins provided a massive boost in signal, which is in particular useful in applications with diluted antigens, such as expansion microscopy or correlative light and electron microscopy (CLEM); (ii) since biotinylation extends to adjacent proteins and is stable over the lifetime of a protein, protein interactions are preserved in time, enabling to monitor dynamic processes in the cell.

## Results

### TurboID-fusion proteins can be accurately localised with fluorescent streptavidin by light microscopy

To establish a robust imaging system for TurboID fusion proteins by streptavidin, we expressed the *T. brucei* nuclear pore proteins NUP110, NUP76, and NUP96 fused to a TurboID-HA tandem tag in procyclic *Trypanosoma brucei* cells using both N and C-terminal fusion for each respective target protein. The tag is composed of the biotin ligase TurboID (*Branon et al., 2018*) and the HA peptide tag. Expression was constitutive from the endogenous locus and labelling relied on the biotin substate present in the medium (0.8 µM in SDM79) (*Moreira do et al., 2023*). Biotinylated proteins were purified by streptavidin affinity and analysed by liquid chromatography coupled to tandem mass spectrometry (LC-MSMS), using both, wild type cells and cells expressing TurboID-GFP for background correction (*Moreira do et al., 2023*). For each bait protein, we quantified and visualized the

biotinylation of all other nuclear pore proteins and known transport factors in *t*-test difference increments (*Figure 1A*, *Figure 1—figure supplement 1*, *Supplementary file 2*). Additionally, we reanalysed previously published data of C-terminally tagged MEX67 and NUP158 in the same way (*Moreira do et al., 2023*, *Figure 1A*, *Figure 1—figure supplement 1*, *Supplementary file 2*). Neither of the bait proteins caused the biotinylation of the entire pore complex. Instead, we observed highly specific labelling patterns for each bait. While the central NUP96 biotinylated most nuclear pore proteins with the marked exception of the nuclear basket, NUP158, and NUP76 caused biotinylation of specific subcomplexes of the pore and MEX67 labelled NUPs lining the channel of the pore (*Moreira do et al., 2023*). The data show that the labelling radius of TurboID, under these conditions, is below the size of the nuclear pore and thus well below the resolution of light microscopy.

Next, we imaged cells expressing nuclear pore proteins fused to the TurboID-HA tandem tag using both cy3-streptavidin and anti-HA in combination with a secondary antibody coupled to the Alexa488 fluorophore (*Figure 1B*). For *T. brucei* NUP132-TurboID-HA, no obvious differences in resolution were observed between antibody-based and streptavidin-based detection (*Figure 1C*). Likewise, for human NUP88, expressed as N-terminal TurboID-4HA fusion protein in HeLa cells, grown in DMEM medium (relying on the biotin of the serum supplement with a low nanomolar concentration), the signal of anti-HA and of streptavidin appeared highly similar (*Figure 1D*). Wild-type cells of both trypanosomes and humans showed only a very low streptavidin signal, indicating that the signal from naturally biotinylated proteins is negligible (*Figure 1—figure supplement 2*). The data demonstrate that imaging of Turbo-ID fusion proteins via streptavidin is a reliable alternative, with a resolution that is indistinguishable from antibody-based immunofluorescence in standard light microscopy, which is in agreement with published data on experimentally determined BioID labeling radii (*Branon et al., 2018*; *Kim et al., 2016*). The one exception is very motile proteins that produce a 'biotinylation trail' distinct to the steady state localisation; these exceptions, and how they can be exploited to understand protein interactions, are discussed in Chapter 4 below.

## TurboID-fusion proteins can be imaged in phase-separated/protein-dense regions

We had previously expressed one of the *T. brucei* nuclear mRNA transporters, *Tb*MEX67, fused to TurboID-HA and observed that anti-HA failed to stain MEX67 at the nuclear pores, indicating a possible failure of the antibody to either penetrate phase-separated areas or to bind the epitope in this environment (details in introduction, *Moreira do et al., 2023*, *Figure 2A*).

To test this hypothesis, we focussed on two additional *bona fide* phase-separated regions: the nucleolus and starvation stress granules. We expressed a C-terminal TurboID-HA fusion of the trypanosome nucleolar GTP-binding protein NOG1 (Tb927.11.3120) (*Billington et al., 2023*). The resulting streptavidin signal was exclusively at the nucleolus, which is tractable as a distinct compartment within the nucleus lacking DAPI fluorescence (*Figure 2B*). In contrast, the HA-signal showed no obvious accumulation in the nucleolus and was extremely weak with a patchy distribution throughout the entire cell body, indicating an unspecific background stain (*Figure 2B*). Thus, just like MEX67, the nucleolar protein NOG1 is detectable by streptavidin, but not by anti-HA immunofluorescence. Next, we starved trypanosome cells that were expressing the established starvation stress granule marker PABP2 (*Fritz et al., 2015*; *Kramer et al., 2013*) fused to TurboID-HA and probed the cells with the streptavidin anti-HA combination. Both anti-HA and streptavidin readily detected starvation stress granules (*Figure 2C*). However, the granules detected by anti-HA were in general less distinct and appeared larger than those detected by streptavidin. Indeed, intensity profiles drawn through Z-stack projections of stress granules were 1.4±0.2 fold wider for anti-HA detection in comparison to streptavidin detection and the corresponding HA profiles frequently exhibited a double-peak (*Figure 2D and E*). The data are consistent with anti-HA preferentially staining the periphery of the granules, while streptavidin diffuses into and binds within these dense, phase-separated particles. A switch of the two fluorophores (Cy3 and Alexa488) gave identical results, controlling for microscopy artefacts as a reason for the differences (*Figure 2—figure supplement 1*). Direct eYFP- fluorescence intensity profiles of granules from cells expressing PABP2-eYFP resembled those of the streptavidin stain, ruling out a preferential granule-peripheral localisation of PABP2 as an explanation (*Figure 2—figure supplement 2*). Thus, anti-HA fails to label three distinct phase-separated structures of the trypanosome cell, the nuclear pore channel,

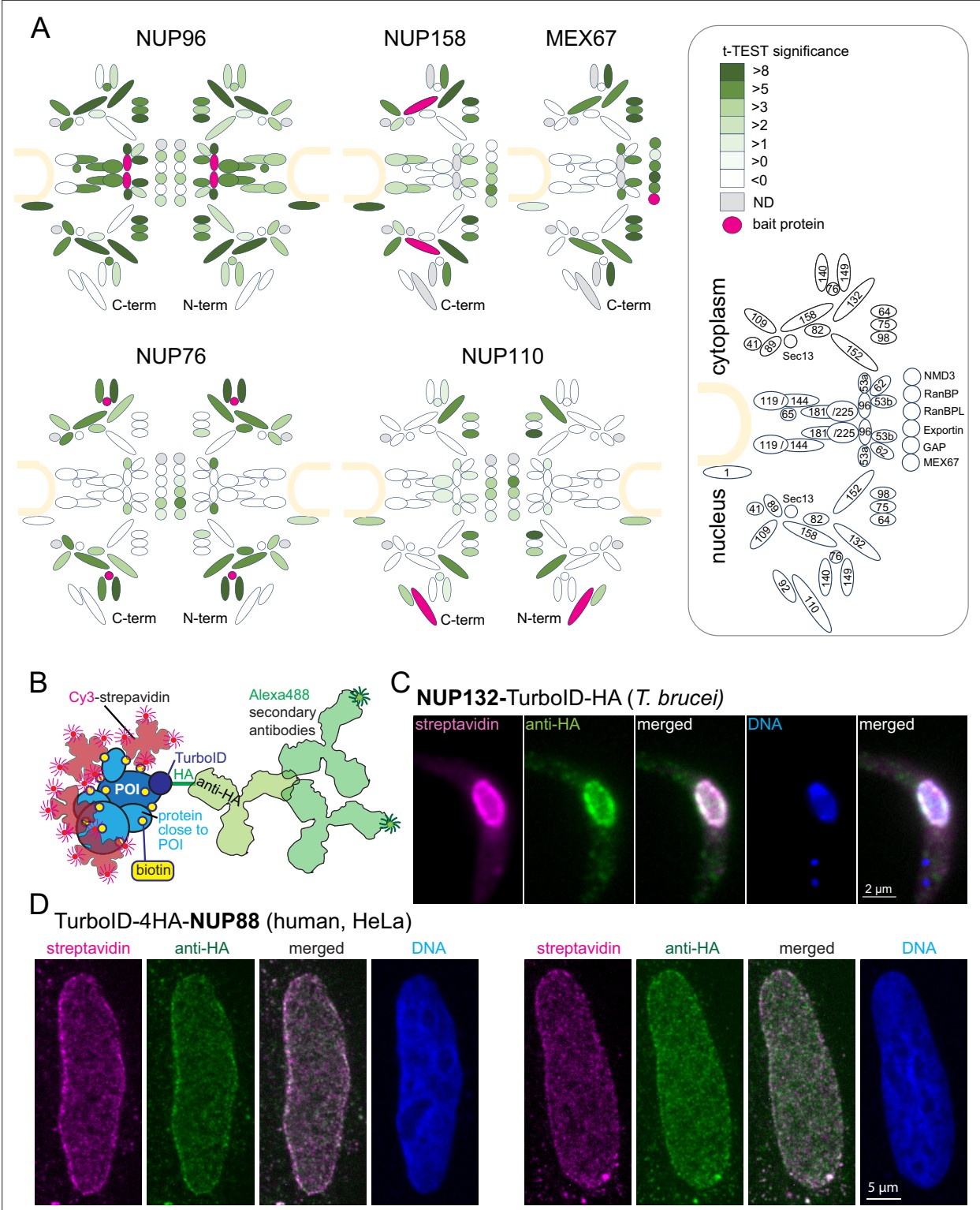

**Figure 1.** The TurboID biotinylation labeling radius is sufficiently small to allow streptavidin-based imaging of target proteins by light microscopy. (**A**) Schematic representation of the trypanosome NPC *Obado et al., 2016* including selected transport factors and their enrichment observed in TurboID experiments followed by streptavidin affinity capture and liquid chromatography coupled to tandem mass spectrometry (LC-MS/MS) analysis. Proteins quantified are filled in shades of green representative of the corresponding *t*-test difference increments. The respective bait protein is drawn in pink and undetected proteins are in grey. NUP96, NUP76, and NUP110 experiments were analysed with TurboID-HA fusions at both respective termini (as indicated). The nuclear envelope is drawn in sand and nucleoporins and transport factors are numbered in the legend (right) according to *Obado et al.,*

*Figure 1 continued on next page*

*Figure 1 continued*

*2016* (NMD3, Tb927.7.970; RanBP1, Tb927.11.3380; RanBPL, Tb927.10.8650; exportin 1, Tb927.11.14340; GAP, Tb927.10.7680),(**B**) Schematics illustrating the imaging concept for a protein of interest (POI) by either Cy3-streptavidin or anti-HA coupled with secondary Alexa488-labelled antibodies. (**C–D**) Nuclear pore proteins of *Trypanosoma brucei* (**C**) and HeLa cells (**D**) were expressed fused to the streptavidin-HA tandem tag and detected with streptavidin-cy3 (pink) and anti-HA (green). Representative single-plane images of a Z-stack is shown, as raw data.

The online version of this article includes the following figure supplement(s) for figure 1:

**Figure supplement 1.** Statistical analysis of NUP TurboID experiments.

**Figure supplement 2.** Streptavidin and anti-HA signal of wild type cells.

the nucleolus, and the inside of starvation stress granules, while streptavidin faithfully decorates all these structures.

The inability of anti-HA to detect target epitopes within phase-separated areas is not restricted to trypanosomes: When expressing TurboID-4HA fusions of the human FG-repeat nucleoporin NUP54 in HeLa cells, streptavidin detected the proteins at the nuclear pores, while no specific anti-HA signal was observed (*Figure 2F*).

Furthermore, the inability to label phase-separated areas is not restricted to anti-HA but applies to other antibodies. We expressed *T. brucei* MEX67 and NOG1 fused to TurboID-Ty1 and failed to label the nuclear pores and the nucleolus with anti-Ty1 (BB2), respectively, while the streptavidin signal displayed the correct localisations (*Figure 2G and H*). Moreover, MEX67 was undetectable at the nuclear pores by polyclonal antiserum (*Figure 2I*; *Pozzi et al., 2023*) and an eGFP-fusion of MEX67 could not be detected by fluorophore-labelled eGFP nanobodies at the pores either (*Figure 2J*). The absence of a nanobody signal rules out that its simply the size of IgGs that prevents the staining of Mex67 at the nuclear pores, as nanobodies are smaller than (tetrameric) streptavidin. In contrast, an eGFP fusion of NOG1 was detectable by eGFP nanobodies at the nucleolus, albeit weakly (*Figure 2J*). In conclusion, most but not all antibodies/nanobodies were ineffective in phase-separated areas.

## Boosting the signal under low-antigen conditions (expansion microscopy, CLEM)

A potential advantage for imaging TurboID fusion proteins with streptavidin is an increased signal: TurboID is expected to cause multiple lysine-biotinylation of both the bait protein and proteins in close proximity, while antibodies only decorate several (polyclonals) or one specific (monoclonals) epitope of a target protein. Indeed, for many TurboID-HA fusion proteins that we analysed by standard light microscopy, the streptavidin signal appeared significantly stronger and devoid of background stain when compared to anti-HA. To quantify this observation, we probed cells expressing NUP158-TurboID-HA with combinations of anti-HA/Alexa488 secondary antibody and Streptavidin-Cy3 (*Figure 3A*) or with Streptavidin-Alexa488 and anti-HA/Alexa594 secondary antibody (*Figure 3B*). Under both labeling conditions, the streptavidin stain resulted in significantly less non-specific cytoplasmic background signal than anti-HA (*Figure 3A–B*). Moreover, the maximum Alexa488 signal of Z-stack projections was 2.9-fold higher for streptavidin than for anti-HA (n=60, unpaired two-tailed *t*-test=9.6E-55) (*Figure 3C*). An increase in fluorescence signal is desirable for the detection of low-abundance proteins, but also for imaging applications that necessitate a reduction in antigen density, such as expansion microscopy or CLEM.

In expansion microscopy, biomolecules are covalently cross-linked to a hydrogel, that is subsequently swollen by osmosis, causing a physical magnification of the sample (*Chen et al., 2015*). Even a relatively small expansion factor of three causes a 27-fold increase in sample volume and thus a 27-fold decrease in antigen-density. The two main expansion microscopy methods are Protein-retention Expansion Microscopy (proExM) (*Tillberg et al., 2016*) and Ultra-structural Expansion Microscopy (U-ExM) (*Gambarotto et al., 2019*). The main difference is that in proExM, antibody labeling is applied prior to the expansion process, while in U-ExM it is done afterward. U-ExM thus generally results in higher resolution, as the linkage-error (difference between the position of the fluorophore of the secondary antibody and the position of the target protein) is not expanded. Next to the decrease in antigen-density as a consequence of the expansion, the signal is further compromised by the harsh conditions associated with the method, as for example an incubation step at 95 °C in alkaline SDS-containing buffer for 90 min for U-ExM and an over-night proteinase K digestion for

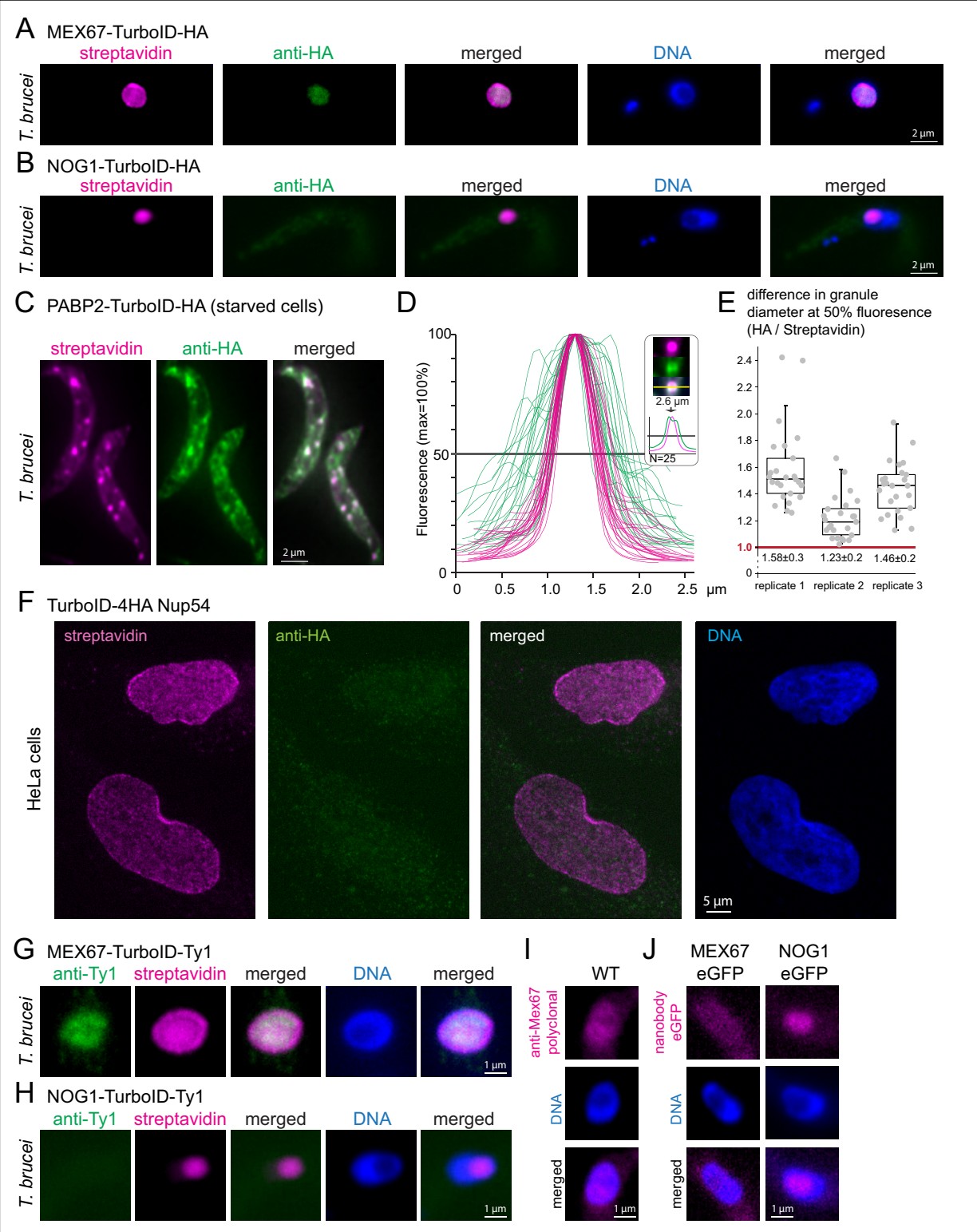

**Figure 2.** Streptavidin can detect targets within phase-separated regions, while most antibodies fail. (**A–B**)*T. brucei* MEX67 (**A**) and *T. brucei* nucleolar protein NOG1 (**B**) were expressed fused to a C-terminal TurboID-HA tandem tag and cells probed with streptavidin-cy3 (pink) and by anti-HA immunofluorescence (green). Representative single plane images of an unprocessed Z-stack series are shown. (**C–E**) Cells expressing the stress granule marker protein PABP2 fused to TurboID-HA were starved (2 hr PBS) and starvation stress granules detected by streptavidin (pink, Cy3) and anti-HA (green, Alexa488). The starvation experiment was performed in biological triplicates. One representative image of starved cells is shown as Z-stack projection (72 slices a 140 nm, sum slices) in C. For each replicate, intensity profiles across one of the larger granules of the cell were measured for

*Figure 2 continued on next page*

*Figure 2 continued*

25 cells in both fluorescence channels. The profiles for replicate 1 are shown in D. For each granule, the granule diameter was calculated from the profiles at 50% fluorescence and the difference in diameter between the HA- and streptavidin stain is presented in E for each replicate, as quotient of granule diameters. Note that despite differences between the three replicates, likely arisen from starvation conditions being not 100% reproducible, the HA stain consistently delivered a larger granule diameter than the streptavidin stain, consistent with preferentially peripheral staining of the granule by anti-HA. For replicate 2, the fluorophores were switched, with essentially the same result (*Figure 2—figure supplement 1*). (**F**) Human NUP54 fused to TurboID-4HA was expressed in HeLa cells and cells were probed with both anti-HA (green, Alexa488) and streptavidin (Cy3, shown in pink). Streptavidin, but not anti-HA detects NUP54 at the nuclear pores. A single plane image of a Z-stack is shown as raw data. (**G** and **H**) *T. brucei* MEX67 (**G**) and NOG1 (**H**) were expressed as TurboID-Ty1 fusion proteins and detected with streptavidin (Cy3, shown in pink) and anti-Ty1 (BB2, green). Representative single plane images of unprocessed Z-stack images are shown. (**I**) Trypanosome wild-type (WT) cells were probed for MEX67 with polyclonal antiserum (kind gift of Mark Carrington, University of Cambridge; secondary antibody Alexa 488, shown in pink). One representative single plane image of an unprocessed Z-stack image is shown.(**J**) *T. brucei* MEX67 (left) and NOG1 (right) were expressed as eGFP fusion proteins and detected with Cy5 labelled eGFP nanobodies. Representative single plane images of unprocessed Z-stack images are shown.

The online version of this article includes the following figure supplement(s) for figure 2:

**Figure supplement 1.** The larger diameter of stress granules of anti-HA stained granules in comparison with streptavidin stained granules is not caused by the different fluorophores.

**Figure supplement 2.** PABP2-eYFP is not at the periphery of stress granules.

ProExM. To assess the potential of streptavidin imaging, we employed both proExM and U-ExM on cell lines expressing TurboID-HA fusions of two trypanosome nuclear-pore localized proteins (NUP76 and MEX67) and compared the signals from streptavidin and anti-HA (*Figure 3D*). For both proteins, streptavidin caused a strong and uniform stain of all nuclear pores with both expansion microscopy methods. In contrast, anti-HA failed to detect nuclear pores either completely (MEX67) or partially (NUP76) in U-ExM. In proExM, pores could be stained evenly with anti-HA, but the signal was much lower in comparison to streptavidin, in particular for NUP76. To our knowledge, this is the first time that streptavidin imaging of a biotin ligase-fused protein was used in combination with expansion microscopy. In our hands, streptavidin clearly outperformed antibodies in both expansion protocols and with both target proteins that we tested, giving a significantly stronger signal with less background.

CLEM causes an even larger reduction in accessible antigens than expansion microscopy. Thin slices of a resin-embedded sample are first used for immunofluorescence and imaged by light microscopy, and, in a second step, imaged by electron microscopy. The resin largely prevents antibody penetration and probing is, therefore, restricted to the small fraction of antigens present at the surface of the resin slice. Trypanosome cells expressing NUP96-TurboID-HA or NUP158-TurboID-HA were embedded in LR-white and slices were probed with streptavidin and anti-HA. For both proteins, the streptavidin signal detected numerous pores for each nucleus, while anti-HA detected between 0 and 2 pores and, in addition, caused considerably higher background signals (*Figure 3E* and *Figure 3—figure supplement 1*). Streptavidin is thus more suitable for CLEM, and an example, targeting NUP158-TurboID-HA with both steps, is shown in *Figure 3F*.

In summary, streptavidin detection causes a massive increase in signal that is in particularly beneficial under low-antigen density conditions, and thus offers significant implications in expansion microscopy and CLEM.

## Visualisation of protein interactions

We found that in most cases, streptavidin labeling faithfully reflects the steady-state localisation of a bait protein, e.g., the localisation resembles those observed with immunofluorescence or direct fluorescence imaging of GFP-fusion proteins. For certain bait proteins, this is not the case, for example, if the bait protein or its interactors have a dynamic localisation to distinct compartments, or if interactions are highly transient. It is thus essential to control streptavidin-based de novo localisation data by either antibody labelling (if possible) or by direct fluorescence of fusion-proteins for each new bait protein. However, any additional signal from biotinylated partner proteins at a different location can also be leveraged to gain relevant information on the function and dynamics of the bait protein. Three respective examples are detailed below:

The first is PABP2, one of two trypanosome poly(A) binding proteins, that has established, unambiguous cytoplasmic localisation, when visualised with either anti-HA or when expressed fused to a fluorescent protein (*Figure 4A–B*; *Billington et al., 2023*; *Kramer et al., 2013*). However, when

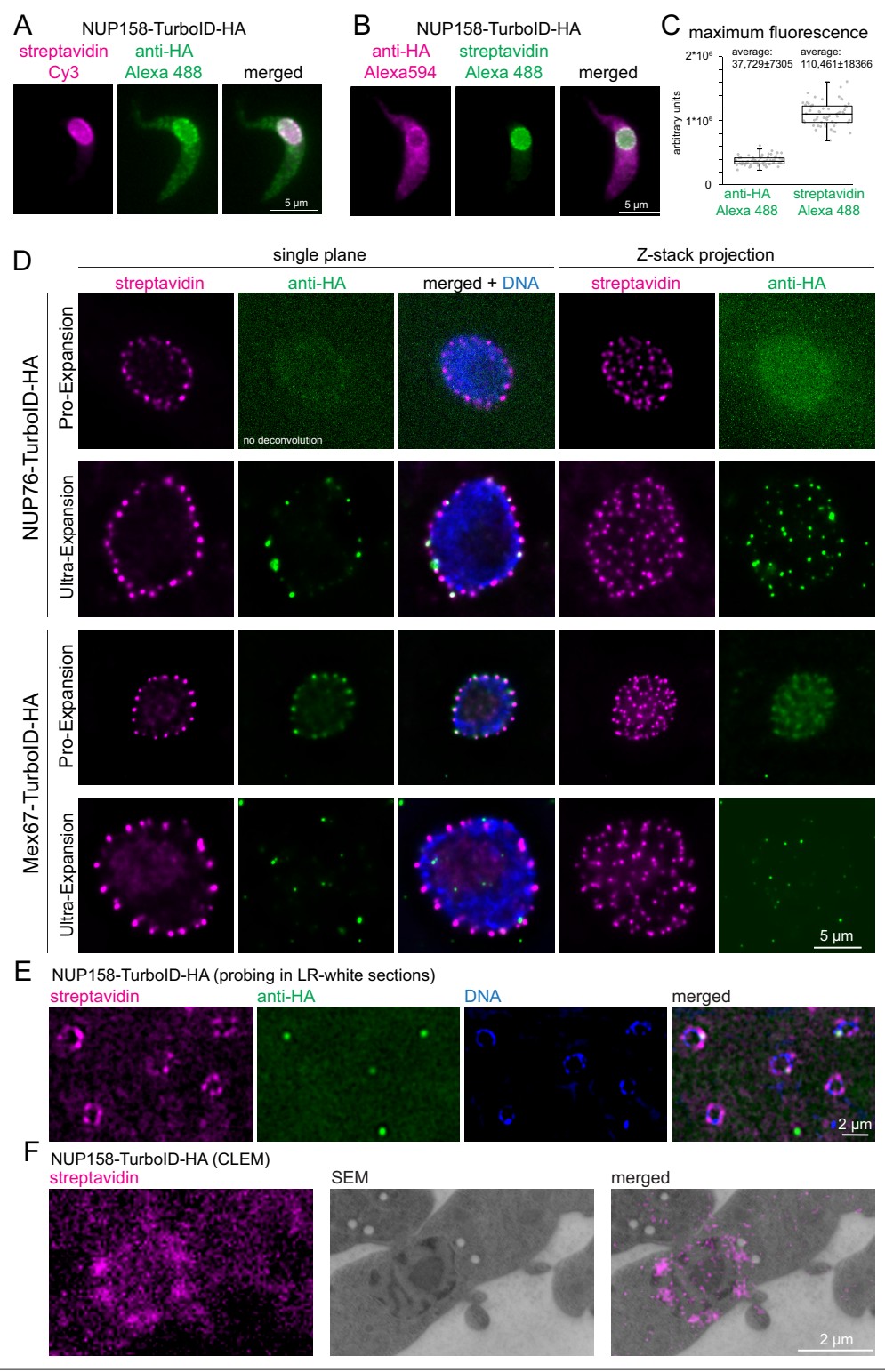

**Figure 3.** Streptavidin imaging yields higher signal intensities than immunofluorescence. (**A–C**) Enhanced signal in standard light microscopy. Trypanosome cells expressing NUP158-TurboID-HA were labelled with combinations of either streptavidin-Cy3 and anti-HA/Alexa488 secondary (**A**) or with streptavidin-Alexa488 and anti-HA//Alexa 594 secondary (**B**) Z-stack images were recorded (48 slices a 140 nm). Representative, unprocessed single plane images are shown (**A** and **B**) The maximum Alexa488 fluorescence was quantified from Z-stack projections (sum slices) from 60 cells probed with anti-HA or streptavidin; the data are presented as a dot blot (waist is median; box is IQR;

*Figure 3 continued on next page*

*Figure 3 continued*

whiskers are ±1.5 IQR) (**C**) (**D**) Improved signal in expansion microscopy. Trypanosome cells expressing NUP76-TurboID-HA or MEX67-TurboID-HA were imaged using Pro-expansion or Ultra-expansion microscopy. Single plane and Z-stack projections (sum slices) of the streptavidin and anti-HA signal are shown for one representative nucleus. All images were deconvolved in proExM, except for NUP76 anti-HA. (**E–F**) Improved signal in correlative electron and light microscopy (CLEM). Trypanosome cells expressing NUP158-TurboID-HA were embedded in LR-White resin. Slices were probed with streptavidin and anti-HA and imaged by light microscopy (**E**) followed by electron microscopy (CLEM) (**F**).

The online version of this article includes the following figure supplement(s) for figure 3:

**Figure supplement 1.** Streptavidin and anti-HA signal on LR-white embedded sections.

PABP2-TurboID-HA expressing cells are probed with streptavidin, a signal at the posterior pole of the cell is observed, in addition to the expected cytoplasmic signal (*Figure 4A*). Localisation to the posterior pole is an exclusive feature of the trypanosome mRNA decapping enzyme ALPH1 and its four interaction partners: a small proportion of this mRNA decapping complex localises to the posterior pole, in a dynamic manner, while the majority remains cytoplasmic (*Kramer, 2017*; *Kramer et al., 2008*; *Kramer et al., 2023*; *Figure 4B*). The streptavidin signal at the posterior pole of the PABP2-TurboID-HA cells thus likely reflects a historic cytoplasmic interaction between PABP2 and the mRNA decapping complex. The existence of this interaction is supported by the identification of all five posterior-pole localised members of the decapping complex by mass spectrometry data from a PABP2 BioID experiment (*Moreira do et al., 2023*). Notably, imaging with streptavidin adds information about protein dynamic interactions, that mass spectrometry data alone cannot uncover. In this case, it shows that the fraction of mRNA decapping proteins at the posterior pole is dynamically exchanged with the fraction in the cytoplasm.

The second example is MLP2 (also called NUP92), a divergent nucleoporin of Kinetoplastida involved in chromosome distribution during mitosis (*Holden et al., 2014*; *Morelle et al., 2015*). MLP2 localisation is cell cycle-dependent: while exhibiting a nuclear localisation during interphase, it migrates to the spindle and spindle pole during mitosis (*Holden et al., 2014*; *Morelle et al., 2015*; *Figure 4—figure supplement 1*). We expressed MLP2 fused to an N-terminal TurboID-HA tag and examined the streptavidin and HA signal in interphase and mitotic cells (*Figure 4C*). Note that in trypanosomes the kinetoplast (mitochondrial DNA, visible as a small dot in DAPI stain) divides prior to the nucleus, and numbers and positions of kinetoplasts and nuclei are established markers for cell cycle stages (*Sherwin and Gull, 1989*). We observed that the streptavidin signal 'lagged behind' the anti-HA signal at all cell cycle stages. This is most obvious in anaphase, when anti-HA exclusively stains the spindle pole body, while streptavidin decorates both, the nuclei and the spindle pole body. Thus, for proteins with cell-cycle dependent localisation, streptavidin-labelling provides information on both, the previous localisation(s) of the bait, in addition to its current position.

The third example is nuclear pore proteins. For all NUPs that we tested, we observed the expected streptavidin signal at the nuclear pores (see below). However, for some NUPs, for example, NUP65, we observed additional cytoplasmic streptavidin stain that is absent for others, as for example NUP75 (*Figure 4D–E*). There were also differences in the amount of nucleoplasmic biotinylation between NUPs. For example, NUP75 showed more nuclear labelling than NUP65 (*Figure 4D–E*). The likeliest explanation is, that some NUPs are contacting and thus proximity-labelling proteins in transit. The streptavidin stain in the nucleoplasm or cytoplasm thus provides information about the functions of the individual NUPs in protein import or export, respectively.

Importantly, tracking of proteins by streptavidin imaging requires orthogonal controls, as the imaging alone does not provide information about the nature of the biotinylated proteins. These can be proximity ligation assay, mass spectrometry, or specific tagging visualisation of protein suspects by fluorescent tags. Once these orthogonal controls are established for a specific tracking, streptavidin imaging is an easy and cheap, and highly versatile method to monitor protein interactions in a specific setting.

## A phase-separation map of the *T. brucei* nuclear pore

We reasoned that our findings that all phase-separated regions of trypanosomes that we tested are inaccessible to anti-HA can be exploited as a tool for de novo identification of phase-separated

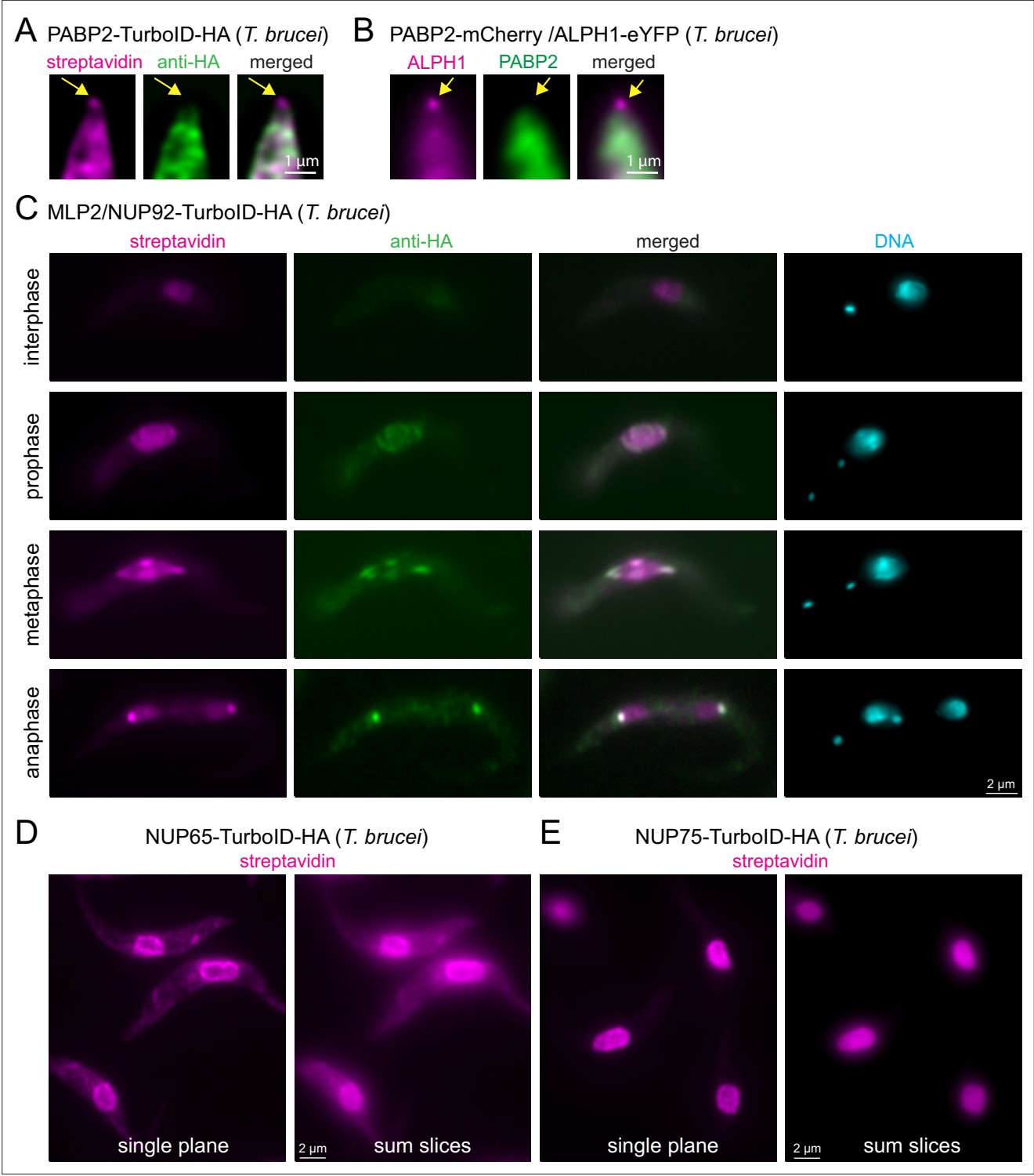

**Figure 4.** Visualisation of protein interactions with TurboID. (**A**) Trypanosome cells expressing PABP2-TurboID-HA were probed with streptavidin and anti-HA. A representative image of the posterior part of a cell is shown (single plane of a Z-stack processed by computational clearing). The position of posterior pole granule is indicated by arrows. (**B**) Trypanosome cells were transformed to co-express PABP2-mChFP and ALPH1-eYFP. A representative image of the posterior part of a cell is shown (single plane of a Z-stack processed by deconvolution). The position of posterior pole granule is indicated by arrows. (**C**) Trypanosome cells expressing TurboID-HA-MLP2 were probed with streptavidin and anti-HA. One representative image of an interphase, prophase, metaphase and an anaphase cell are shown. All images are unprocessed single plane images, with the exception of the DAPI image that is a Z-stack projection (max intensity of 48 slices a 140 nm). (**D** and **E**) Trypanosome cells expressing NUP65 (**D**) or NUP75 (**E**) fused to a C-terminal TurboID-HA tag were probed with streptavidin. Single plane and Z-stack projection (sum slices of 48 slices a 140 m) of unprocessed images are shown.

*Figure 4 continued on next page*

*Figure 4 continued*

The online version of this article includes the following figure supplement(s) for figure 4:

**Figure supplement 1.** MLP2 does largely not localise to nuclear pores, but to the nucleus and to the spindle pole.

regions. As a proof of principle, we employed streptavidin-imaging to produce a map of antibody-inaccessible regions of the entire trypanosome nuclear pore complex. We endogenously expressed all known trypanosome nuclear pore proteins fused to TurboID-HA from one allele. For each protein, two cell lines were constructed, one with TurboID-HA fused to the C-terminus, and one with the tag at the N-terminus. We excluded Mlp2/Nup92, as we found this protein localizing to the nucleus rather than the nuclear pores in most interphase cells (*Figure 4—figure supplement 1* and *Figure 4C*), in agreement with (*Morelle et al., 2015*). For two C-terminal fusion candidate proteins (Sec13, NUP140) and one N-terminal (NUP82), we repeatedly failed to obtain cell lines. The remaining 45 cell lines were controlled by western blot and probed with anti-HA to confirm the correct size of the fusion protein (*Figure 5—figure supplement 1*). Next, cells from all cell lines were probed with anti-HA and fluorescent streptavidin. All cell lines showed streptavidin signals predominantly at the nuclear pores, proving the correct localisation of the fusion proteins (*Figure 5A*). For N-terminally tagged NUP64 we also observed between one and two streptavidin dots in the nucleus, consistent with the localisation of eYFP-NUP64 in life cells (*Billington et al., 2023*); a likely mislocalisation.

When evaluating the HA signal at the nuclear pores, we observed no antibody stain for the NUPs lining the central channel, while most NUPs at the periphery of the channel were accessible to the antibody (*Figure 5*). Importantly, there is no correlation between protein abundance, as determined by proteomics (*Tinti and Ferguson, 2022*), and immunofluorescence signal, ruling out low antigen abundance as the reason for its absence (*Figure 5—figure supplement 2*). All the FG-NUPs of the inner channel (which are known to phase-separate) were among the non-accessible proteins, supporting the idea that the crucial determinant preventing anti-HA detection is mainly or entirely phase separation. Likewise, the mRNA-interacting NUP76 complex consisting of NUP76 and the FG NUPs NUP140 and NUP149 appear non-accessible to anti-HA, except when NUP149 is tagged at the N-terminus. The NUP76 interacting protein NUP158 was inaccessible to antibodies when tagged at the N-terminus, but not at the C-terminus: this is consistent with NUP158 being partially phase-separated as it contains FG-repeats exclusively at the N-terminal half. Among the antibody-inaccessible proteins were also some non-FG NUPs. These include proteins of the inner ring, NUP181, NUP144, C-terminally tagged NUP225, and N-terminally tagged NUP65, as well as N-terminally tagged proteins of the outer ring, NUP89 and NUP41. Whether the respective termini of these proteins extend into phase-separated areas, or whether antibody access is prevented by steric or other hindrances remains unknown.

In conclusion, we here provide an antibody access map of the entire *T. brucei* nuclear pore that includes all bone-fine FG-NUPs in addition to some non-FG NUPs.

## Discussion

We here show that the use of the TurboID-HA tandem tag offers an 'all in one' solution to antibody-accessibility problems, without compromising resolution or the strength of the signal. Quite the contrary, streptavidin imaging of TurboID-HA tagged proteins facilitates a massive boost in signal, extending the benefits of the method to protein targets with low antigen abundance. Furthermore, the TurboID-HA tag is suitable for probing protein interactions and provides some evidence for phase-separation.

Like many methods that are frequently used in cell- and molecular biology, streptavidin imaging is based on the expression of a genetically engineered fusion protein: it is essential to validate both, the function and localisation of the TurboID-HA tagged protein by orthogonal methods. If the fusion protein is non-functional or mis-localised, tagging at the other end may help, but if not, this protein cannot be imaged by streptavidin imaging. Likewise, target organisms not amenable to genetic manipulation, or those with restricted genetic tools, are not or less suitable for this method.

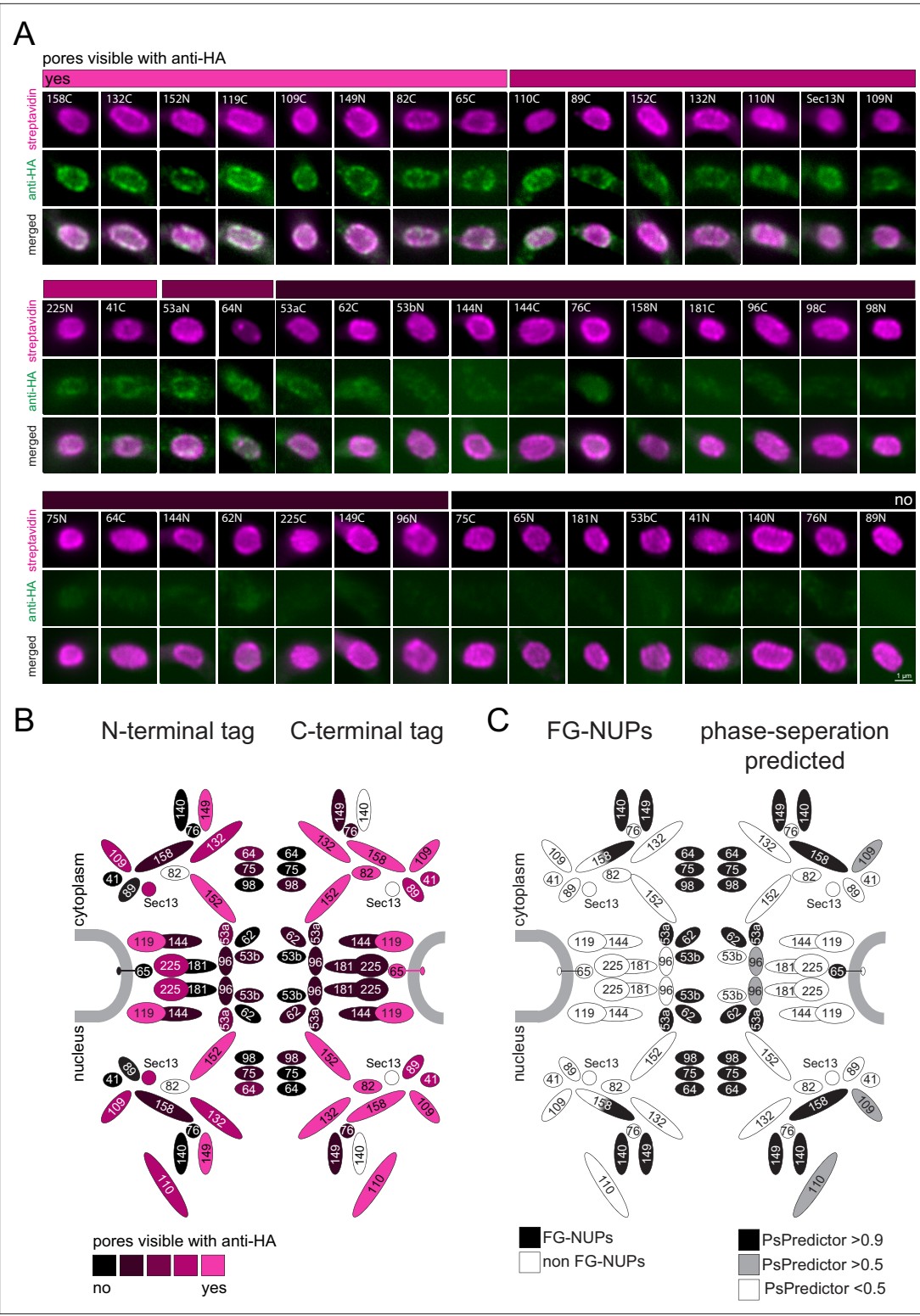

**Figure 5.** A refined map of the *T. brucei* nuclear pore complex. Each known *T. brucei* nuclear pore protein was expressed fused to TurboID-HA, each both at the N- and C-terminus. Cells were labelled with anti-HA and with streptavidin. (**A**) For each NUP, a representative image of the nucleus is shown as unprocessed, single plane of a Z-stack. We evaluated the extent of anti-HA stain in colour increments (shown as a bar above images). (**B**) The HA-signal at the nuclear pores was mapped onto a schematic representation of the trypanosome NPC (modified from ***Obado et al., 2016***) using the same colour increments. (**C**) Scheme of the *T. brucei* nuclear pore, with known FG NUPs (***Obado et al., 2016***) shown in black on the left side and prediction of phase separation (***Chu et al., 2022***) mapped on the right.

*Figure 5 continued on next page*

*Figure 5 continued*

The online version of this article includes the following source data and figure supplement(s) for figure 5:

**Figure supplement 1.** Western blots.

**Figure supplement 1—source data 1.** Uncropped and labelled gels for *Figure 5—figure supplement 1*.

**Figure supplement 1—source data 2.** Raw unedited gels for *Figure 5—figure supplement 1*.

**Figure supplement 2.** The strength of the immunofluorescence signal of HA-tagged nucleoporins does not correlate to estimated protein abundances.

## One protocol suits all - streptavidin readily traces biotin proximity labels

Many proteins are not accessible to antibodies in standard immunofluorescence protocols, in particular proteins localised in phase-separated regions. MEX67 could not be detected at the trypanosome nuclear pore by anti-HA, anti-Ty1, anti-Protein A, polyclonal MEX67 antibodies and even nanobodies to mNeonGreen (this work, *Dostalova et al., 2013*; *Moreira do et al., 2023*; *Pozzi et al., 2023*); likewise, the human Mex67 homologue TAP/Nxf1 evaded detection by antiserum (*Bear et al., 1999*). HA-fused trypanosome FG-NUPs and the human FG-NUP NUP54 could not be decorated with anti-HA. Trypanosome starvation stress granules could only be labelled at the periphery and nucleolar proteins evaded antibody detection in trypanosomes (*Figure 2*) and in human cells (*Misteli, 2008*; *Musinova et al., 2011*; *Sheval et al., 2005*; *Zatsepina et al., 1997*). There are several approaches available to trouble-shoot antigen-accessibility problems (*Piña et al., 2022*). Pre-extraction steps prior to fixation employing Triton X100, high salt or nucleases proved successful for the detection of mammalian Mex67 homologue TAP/NXF1 (*Ben-Yishay et al., 2019*; *Katahira et al., 1999*). Antibody penetration to nucleolar proteins was improved by including protease treatment (*Svistunova et al., 2012*). Furthermore, fixation by cold methanol instead of paraformaldehyde was reported to increase antibody access to proteins (*Neuhaus et al., 1998*). However, the search for individual solutions for each problematic antibody/antigen pair is cumbersome and an improvement of antibody access typically comes at the expense of signal loss and/or disruption of cellular morphology (*Piña et al., 2022*). With streptavidin labeling we present a method that works for every protein (we tested ~30), without compromising cellular morphology and, most importantly, provides a strong signal as significant additional benefit.

In standard light microscopy, streptavidin imaging offers a resolution comparable to labelling with antibodies. The labelling radius of the biotin ligase BirA was estimated to be approximately 10 nm by using the NUP107-160 Y-complex of the mammalian nuclear pore as a molecular ruler (*Kim et al., 2014*), enabling, for example, the determination of substructures of P-bodies and stress granules (*Youn et al., 2018*). It is thus within the range of an average-sized protein and well below the resolution limit of light microscopy. Consistently, cells expressing nuclear membrane-associated proteins fused to BirA (or variants as the related BioID2) can be probed by fluorescent streptavidin with resolutions that are undistinguishable from probing with the respective IgG antibodies (*Kim et al., 2016*; *May et al., 2020*). For the biotin ligase variant TurboID, which has a massively decreased labelling time, the biotin labelling radius was measured to be in a similar (*Branon et al., 2018*; *Kim et al., 2016*) or slightly larger range (*May et al., 2020*). Importantly, the addition of 50 µM external biotin massively increased the labelling radius, while in the absence of excessive biotin, the labelling radius remained comparable to the labelling radius of BirA (*May et al., 2020*). In this work, we avoided the addition of additional biotin to the trypanosome medium (the medium contains 0.8 µM biotin) and serum biotin (approximately 2 nM in mammalian serum *Luong and Vashist, 2020*) was the sole source in HeLa cell medium. Our mass spectrometry data of streptavidin-affinity purified proteins with five nuclear pore-localised TurboID-fusion proteins demonstrates that the labeling radius is well below the size of the nuclear pore complex under these conditions, and thus in agreement with published data. Even in expansion microscopy, with 3.6–4.2-fold expansion factors, we did not notice any decrease in resolution; in fact, we could resolve NUPs that reside at different positions within the pore (e.g. inner ring vs basket) by labeling one with streptavidin and the other with an antibody and thus obtained sub-NUP imaging resolution (our own unpublished data). Even though, theoretically, BioID per se causes a decrease in resolution as biotinylation extends to proteins in close proximity to the bait, this may well be compensated by the lower linkage error when imaging directly via the small, compact-shaped streptavidin-tetramer rather than by a tandem assembly of two large IgG antibodies (*Figure 1B*).

## Streptavidin tracing of biotin proximity labels delivers high signal intensities

One major benefit of streptavidin imaging was the increase in signal compared to immunofluorescence, caused by multiple biotinylation sites of the bait and adjacent proteins serving as an enhancer. An increased signal is beneficial for almost all protein-imaging applications, but in particular for low abundance targets. Most strikingly, streptavidin imaging facilitates imaging targets with low densities as a consequence of the imaging method, as in expansion microscopy and CLEM, and is applicable to all target proteins that tolerate tag fusion to either terminus. Fluorescent streptavidin has been previously used in expansion microscopy to detect biotin residues in target proteins produced by click chemistry (*Sun et al., 2021*). However, to the best of our knowledge, this is the first report that employs fluorescent streptavidin as a signal enhancer in expansion microscopy and CLEM, by combining it with multiple biotinylation sites added by a biotin ligase. Importantly, for both CLEM and expansion, streptavidin imaging is the only alternative approach to immunofluorescence, as denaturing conditions associated with these methods rule out direct imaging of fluorescent tags.

## A combination tag as a probe for phase-separated areas

A further important application of the TurboID-HA tandem tag is the de novo prediction of phase-separated areas in cells. The importance of compartmentalisation by liquid-liquid phase separation for almost all cellular functions, including DNA and mRNA metabolism (*Dai and Yang, 2024*), cell division (*Ong and Torres, 2020*), and development (*So et al., 2021*) becomes increasingly evident. Aberrant phase separation can cause disease, as for example neurodegenerative disorders (*Chakraborty and Zweckstetter, 2023*; *Tsoi et al., 2023*), underlining the relevance of novel tools for the identification and characterisation of phase-separated regions. For individual proteins, phase separation can be predicted in silico using machine learning, with limited, but increasing accuracy (*Venko and Žerovnik, 2023*). For a defined mixture of molecules, phase separation can be tested in vitro (*Alberti et al., 2018*; *Zhang and Shen, 2023*). However, methods to test phase separation in the cellular context are scarce. The correlation between antibody-inaccessibility and phase separation, that we observed in this study, offers the opportunity to use the TurboID-HA tag to probe for phase separation de novo. Notably, the streptavidin stain concomitantly provides a convenient intrinsic control to confirm the correct localisation of the fusion protein. Moreover, proteins with complex localisation patterns can be simultaneously detected in both, phase-separated and non-phase-separated regions, as we demonstrated for tracing MEX67 to both, the nucleoplasm and nuclear pores (*Figure 1A and G*). In trypanosomes, all *bona fide* phase-separated regions were inaccessible to anti-HA, namely the nucleolus, starvation stress granules, and the FG NUP environment. The data strongly suggest that the absence of an HA immunofluorescence signal correlates with phase-separation. However, we cannot rule out the possibility, that some proteins may be inaccessible to anti-HA for different reasons, as for example steric hindrance. The latter scenario would equally favor streptavidin over IgG antibodies for target protein binding due to its much smaller dimensions and higher binding affinity. We observed some non-FG Nups to be anti-HA inaccessible, which were mostly NUPs of the inner ring, close to the core of FG NUPs: whether these NUPs localise to phase-separated areas is not known. Importantly, problems to detect phase-separated proteins by antibodies appear conserved across species (*Bear et al., 1999*; *Misteli, 2008*; *Musinova et al., 2011*; *Sheval et al., 2005*; *Zatsepina et al., 1997*). We show data for the human FG-NUP Nup54 as an example of an anti-HA-inaccessible protein (*Figure 2F*) as proof of principle, that the de novo identification of phase-separation is applicable to human cells.

Why are most antibodies and, to some extent, even fluorophore conjugated nanobodies, prevented from labelling proteins in phase-separated environments, while streptavidin never fails? We cannot distinguish whether the lack of labelling is caused by an access problem of the antibody penetrating the phase-separated area by a reduced binding affinity between antibody and antigen in phase-separated environments, or by a combination of both. Still, the smaller size of streptavidin in comparison to an IgG tandem-antibody pair (*Figure 1b*) likely favors its access into dense regions. Furthermore, IgG antibodies as well as single-chain antibodies typically have affinities in the nanomolar range, compared to the femtomolar affinity of the biotin-streptavidin interaction, which is indeed among the strongest non-covalent interactions in nature (*Chivers et al., 2011*). Together with the extremely high structural stability of streptavidin towards temperature, pH, and denaturing conditions

(*Laitinen et al., 2006*) this can explain the ability of this probe to potently outperform antibodies in phase-separated environments.

## Proximity-labelling reports a history of dynamic protein interactions

Lastly, we propose streptavidin imaging as a novel tool to monitor dynamic interactions between proteins. Since BioID proximity labeling is not restricted to the bait protein, but extends to all interacting partners, the history of interactions is preserved. Hence, changes in protein sub-complexes can be imaged over time. We provide three examples, covering protein complex dynamics in cell cycle regulation and mRNA metabolism as well as nuclear transport. There are many other potential applications. While we relied on an endogenous, constitutive expression of TurboID fusion proteins, a tightly controlled, inducible system, as for example, Split-TurboID (*Cho et al., 2020*) in combination with light activation (*Chen et al., 2022*; *Shafraz et al., 2023*) would allow pulse-chase experiment, delivering quantitative data on the kinetics of processes.

Altogether, streptavidin imaging is a highly versatile tool with multiple applications, ranging from a simple boost in a signal that can be leveraged to increase imaging sensitivity in expansion microscopy and CLEM, to the ability to probe for phase-separated areas and to monitor dynamic protein interactions. It is thus an important addition to the available imaging toolbox.

## Materials and methods

### Culture and genetic modification of trypanosomes

*T. b. brucei* Lister 427 procyclic cells were cultured in SDM-79 (*Brun, 1979*). Generation of transgenic trypanosomes was performed using standard methods (*McCulloch et al., 2004*). Almost all trypanosome fusion proteins were expressed from the endogenous locus, by modifying one allele via a PCR-based approach reliant on a (modified) version of the pPOT vector system (*Dean et al., 2015*). Occasionally, a plasmid-based system was used for endogenous tagging (*Kelly et al., 2007*). For overexpression of ALPH1-eYFP, a tetracycline inducible system was used (*Sunter et al., 2012*) and expression was induced for 24 hr. Details about all fusion constructs are provided in *Supplementary file 1*. The TurboID-HA tandem tag encodes the TurboID open reading frame, followed by a two-amino acid long spacer (AS), followed by the HA tag (YPYDVPDYA). The TurboID-Ty1 tandem tag encodes the TurboID open reading frame, followed by a short linker (ASGSGS), followed by the Ty1 tag (EVHTNQDPLD).

### Culture and genetic manipulation of HeLa cells

pEGFP-N1 was modified to contain the sequence encoding TurboID-4HA with a stop-codon upstream of the eGFP sequence. Sequences of the open reading frames of NUP88 and NUP54 were amplified from human cDNA and cloned in frames downstream of the TurboID-4HA sequence. This results in the expression of NUP88 or NUP54 fused to the N-terminal TurboID-4HA tag, without the eGFP. The sequences of the fusion proteins are provided in *Supplementary file 1*. HeLa cells were grown in Dulbecco's modified Eagle's medium (DMEM; Gibco by Life Technologies, Darmstadt, Germany) supplemented with 10% fetal calf serum (FCS; Capricorn Scientific, Ebsdorfergrund, Germany) and 1% penicillin-streptomycin (Thermo Fisher Scientific, Dreireich, Germany) at 37 °C and 5% $CO_2$. Cells were transfected with the respective TurboID-HA fusion constructs using Effectene (Qiagen, Hilden, Germany) following the manufacturer's protocol and incubated for ~24 hr in a petri dish containing a coverslip. Immunofluorescence/streptavidin labeling was performed on cells attached to the coverslips, as detailed below for trypanosomes.

### Immunofluorescence and streptavidin labelling

About $1\times10^7$ procyclic-form *T. brucei* cells, harvested at a density of $5\times10^6$ cells/ml, were washed once in 1 ml SDM79 without hemin and serum and resuspended in 500 µl PBS. For fixation, 500 µl of 8% paraformaldehyde was added for 20 min while rotating. After the addition of 7 ml PBS supplemented with 20 mM glycine, cells were pelleted, resuspended in 150 µl PBS, and spread on poly-lysine-coated slides (within circles drawn with a hydrophobic pen). Cells were allowed to settle for 15 min, before removing surplus PBS and permeabilizing cells with 0.5% Triton X-100 in PBS. Slides were rinsed in PBS, and cells were then blocked in 3% bovine serum albumin (BSA) in PBS for 30 min, followed by 60 min

incubation with rabbit mAb-anti-HA C29F4 (1:500 dilution; Cell Signaling Technology) and Streptavidin-Cy3 (1:200 dilution; Jackson Laboratories) in PBS/3% BSA. Slides were washed in PBS (three times for 5 min), then incubated with anti-rabbit Alexa Fluor Plus 488 (1:500 dilution, A32731 Invitrogen) in PBS/3% BSA for 50 min and a further 10 min upon addition of 4',6-diamidino-2-phenylindole (DAPI) (0.1 µg/ml). Slides were washed 3×5 min in PBS and embedded into ProLong Diamond Antifade Mountant (Thermo Fisher Scientific). For control experiments, the following antibodies/nanobodies were used: Ty1-tag monoclonal antibody BB2 (hybridoma supernatant, 1:300); *T. brucei* MEX67 polyclonal antiserum (1:1000, kind gift of Mark Carrington); and anti-GFP nanobody (GFP-Booster Alexa Fluor 647 Chromotek gb2AF647, 1:1000).

## Microscopy

For most fluorescence microscopy experiments, images were acquired using a DMI8 widefield microscope (Thunder Imager, Leica Microsystems, Germany) with an HCX PL APO CS objective (100 x, NA = 1.4, Leica Microsystems) and Type F Immersion Oil (refractive index = 1.518, Leica Microsystems). The microscope was controlled using LAS-X software (Leica Microsystems). Samples were illuminated with an LED8 light source (Leica). Excitation light was selected by using the following filter sets: Ex 391/32 nm; DC 415 nm; Em 435/30 nm (DAPI), 436/28 nm; DC 459 nm; Em 519/25 nm (AlexaFluor 488), 554/24 nm; DC 572 nm; Em 594/32 nm (Cy3), 578/24 nm; DC 598 nm; Em 641/78 nm (AlexaFluor 594). Images were captured using a K5 sCMOS camera (Leica, 6.5 µm pixel size). Between 50 and 70 images in 140 nm distances were recorded for each Z-stack. Exposure times were 100–200 ms for DAPI and between 200–800 ms for all other fluorophores. For some images, an inverted wide-field microscope Leica DMI6000B (Leica Microsystems GmbH, Wetzlar, Germany) equipped with a 100 x oil immersion objective (NA 1.4) and a Leica DFC365 camera (6.5 µm/pixel) was used. Filter sets were (i) ex: 340–380 nm, dc: 400 nm, em: 450–490 nm (DAPI), (ii) ex: 450–490 nm, dc: 495 nm, em: 500–550 nm (eYFP), (iii) ex: 530–560 nm, dc: 570 nm, em: 572–648 nm (TMR). Image acquisition was done like on the thunder imager. Images are shown either unprocessed (raw data) or processed by instant computational clearing (Software provided by the Thunder imager) or deconvolution using Huygens Essential software (Scientific Volume Imaging BV). Presentations are either as a single plane or as Z-stack projection (sum slices), as indicated. Scanning electron microscopy and correlation with light microscopy was done as previously described (*Kramer et al., 2021*).

## Affinity enrichment of biotinylated proteins and on beads digestion

The isolation of biotinylated proteins from trypanosomes and preparation for mass spectrometry was done as previously described (*Moreira do et al., 2023*), except that 1 mM biotin was added during the elution step from the streptavidin beads by tryptic digest, to improve recovery of biotinylated peptides.

## Western blots

Western blotting was performed using standard methods. Detection of biotinylated proteins was done with Streptavidin-IRDye680LT (LI-COR) (1:20,000). Proteins with HA-tag(s) were detected using rat mAb-anti-HA 3F10 (Roche, 1:1000) as the primary antibody and IRDye 800CW Goat anti-Rat IgG (Licor, 1:20,000) as the secondary antibody.

## Mass spectrometry and proteomics analysis

TurboID-eluted peptides were resuspended in 50 mM $NH_4HCO_3$ and passed over C18 stage tip columns as described (*Rappsilber et al., 2007*) and then analysed by LC–MS/MS on an Ultimate3000 nano rapid separation LC system (Dionex) coupled to an LTQ Orbitrap Fusion mass spectrometer (Thermo Fisher Scientific). Minimum peptide length was set at seven amino acids allowing a maximum of two missed cleavages, and FDRs of 0.01 were calculated at the levels of peptides, proteins, and modification sites based on the number of hits against the reversed sequence database. Spectra were processed using the intensity-based label-free quantification (LFQ) in MaxQuant version 2.1.3.0 (*Cox and Mann, 2008*; *Cox et al., 2014*) searching the *T. brucei* strain TREU927 annotated protein database (version 63) from TriTrypDB (*Aslett et al., 2010*). Statistical analyses of LFQ data were done in Perseus (*Tyanova et al., 2016*). LFQ values were filtered for at least one valid value in the bait sample group, log2 transformed and missing values imputed from a normal distribution of intensities

around the detection limit of the mass spectrometer. These values were subjected to a Student's t-test comparing the respective bait sample groups to an untagged control (wt parental cells) triplicate sample group. The resulting t-test difference is the difference between the means of logarithmic abundances of a protein group. Bait samples were prepared in duplicate (NUP76, NUP96, NUP110) or triplicate (MEX67, NUP158). All proteomics data have been deposited at the ProteomeXchange Consortium via the PRIDE partner repository (*Perez-Riverol et al., 2019*) with the dataset identifier PXD031245 (MEX67, NUP158, wt control, GFP control) and PXD047268 (NUP76, NUP96, NUP110).

## Ultrastructural expansion microscopy (U-ExM) with procyclic *T. brucei*

The U-ExM method was performed as described in *Gambarotto et al., 2021* with some modifications. $1 \times 10^7$ trypanosome cells were harvested (1500 g, 10 min) and the pellet was resuspended in 500 µl PBS. 500 µl 8% formaldehyde (FA, 36.5–38%, F8775, Sigma)/8% acrylamide (AA, 40%, A4058, Sigma) diluted in PBS was added to achieve a final concentration of 4%FA/4%AA. Cells were allowed to settle on a 12 mm round coverslip treated with poly-L-lysine (P8920, Sigma) placed in a well of a 24-well plate. The adjacent wells were filled with ddH$_2$O, the plate sealed with parafilm, and incubated at 37 °C for 5 hr. A small piece of parafilm was placed on top of a humid chamber, where a drop of monomer solution (sodium acrylate 23% (w/v) (408220, Sigma), AA 10% (w/v), N-N'-methylenebisacrylamide 0.1% (w/v) (M7279, Sigma) in PBS) was added. The coverslip was placed on top of the drop, with the cells facing down. The coverslip was removed, cleaned from excess liquid with a laboratory wipe, and placed in 40 µl of monomer solution freshly complemented with 1 µl 10% APS (327085000, Acros Organic) and 1 µl 10% TEMED (T7024, Sigma). Sodium acrylate stock solution 38% (w/v) was prepared by dissolving 1.9 g sodium acrylate in 3.1 ml of ddH$_2$O followed by filtering through a 0.2 µm filter. The humid chamber was closed, and the gel was allowed to polymerize for 1 hr at 37 °C. Coverslips were transferred to a well of a 6-well plate containing 1 ml denaturation buffer (200 mM SDS (A1112, Applichem), 200 mM NaCl (A2942, Applichem) and Tris-HCl pH 9 (5429.3, Roth)) and incubated for 15 min with gentle shaking. Gels were detached from the coverslips with a spatula and transferred to a 1.5 mL Eppendorf tube. The tube was filled with denaturation buffer to the top, closed, and incubated at 95 °C for 90 min in a heating block. Gels were transferred to a beaker with 100 ml ddH$_2$O for the first round of expansion for 30 min. Water was exchanged and two more rounds of expansion were done, the last one being overnight at room temperature. The next day, gels were carefully transferred to a Petri dish and imaged to measure the expansion factor. Gels were transferred to a fresh beaker containing 100 ml PBS and incubated for 15 min. This was repeated once and then the gel was trimmed to a square of about 12×12 mm which was transferred to a 24-well plate for antibody incubation. Gels were incubated with 0.5 ml PBS BSA 3% (A1391, Applichem) with primary antibodies (mouse-anti-HA monoclonal antibodies (C29F4, Cell Signaling) at 1:500) at 37 °C for 3 hr in a plate shaker at 1500 RPM. Gels were transferred to a six-well plate for washing. They were washed three times with 10 ml PBS-T (0.1% Tween20 in PBS) for 10 min with agitation at room temperature. Gels were transferred back to a 24-well plate and incubated with 0.5 ml PBS BSA 3% with secondary antibodies AlexaFluor 488 goat anti-rabbit (A11008, Sigma, 1:500), Cy3-streptavidin and 0.1 µg/ml DAPI (D9542, Sigma) at 37 °C for 3 hr in a plate shaker at 1500 RPM. Gels were transferred to the 6-well plate and washed four times with 10 ml PBS-T for 10 min with agitation at room temperature, all steps from now on are protected from light. Gel pieces were expanded again in 100 ml ddH$_2$O in a beaker, with three water exchanges every 30 min, and the last expansion step was done overnight at room temperature. An imaging chamber Nunc Lab-Tek II with one chamber (734–2055, Thermofisher Scientific) was covered with a layer of poly-D-lysine (A3890401, Gibco) for 1 hr and left to dry overnight. For imaging, the gel pieces were trimmed to fit the chamber, excess water was removed with a laboratory wipe and the gel was placed in the imaging chamber, taking care not to shift after the initial placement. A drop of water was added to cover the top of the gel. The gel chamber was closed with the lid to avoid evaporation during imaging. For longer imaging sessions, drops of water were periodically added to prevent gel drying. Since cells can be on either side of the gel, the gel was divided into two pieces, one facing up and one facing down, to ensure cells can be detected, on either piece.

## Protein retention expansion microscopy (proExM) in procyclic *T. brucei*

The proExM method was performed as described in *Tillberg et al., 2016* with some modifications. $5 \times 10^7$ cells were harvested by centrifugation at 1500 g for 10 min, washed once in 10 ml PBS and

finally resuspended in 1 ml of PBS. 1 ml of PFA 8% (P6148, Sigma) was added and cells were fixed for 20 min at room temperature with rotation. 13 ml 20 mM glycine (50046, Sigma) in PBS (freshly prepared from a 2 M stock) was added to quench and cells were harvested at 1500 g for 10 min. Cells were resuspended in 1 ml PBS and let to adhere to a 12 mm coverslip, previously treated with poly-L-lysine, placed in a well of a 24-well plate for 15 min. The well was rinsed with PBS and the coverslip was incubated with PBS 0.5% Triton for 5 min for permeabilisation. Wells were rinsed again with PBS and coverslips were incubated with PBS 3% BSA for 30 min for blocking. 50 µl of primary antibody solution (rabbit anti-HA monoclonal antibodies (C29F4, Cell Signaling) 1:50 diluted in PBS BSA 1%) were placed on a piece of parafilm in a humid chamber and the cover slip was placed on the top (cells facing down) for 1 hr in a humid chamber. The coverslip was returned to the well of the 24 well plates, cells facing up, and wells were washed with PBS (3×5 min) with agitation. 250 µl secondary antibodies (streptavidin-Cy3 (SA1010, Sigma) 1:1000; AlexaFluor 488 goat anti-rabbit (A11008, Sigma) 1:250) and DAPI ((D9542, Sigma) 1:1000) were added, diluted in 250 µl PBS BSA 1%. Wells were washed with PBS (4×5 min). From now on, all steps were done protected from light. To anchor the proteins to the gel, we prepared a solution of 5 mg AcX (Acryloyl-X, SE, 6-((acryloyl)amino)hexanoic Acid, Succinimidyl Ester, A20770, Invitrogen) in 500 µl anhydrous DMSO (D12345, Invitrogen); AcX solution was opened and aliquoted in a desiccated environment at – 20 °C. AcX was diluted 1:100 in PBS, a drop was placed on a piece of parafilm in a humid chamber, and the coverslips were placed top-down into the AcX drop and incubated at room temperature for 16 hr. Coverslips were returned to a well of a 24-well plate, washed twice for 5 min with PBS, and stored in the fridge until the end of the day.

Monomer solution was prepared with sodium acrylate 8.6% (w/v) (408220, Sigma), AA 2.5% (w/v), N-N'-methylenebisacrylamide 1.5% (w/v) (M7279, Sigma) and NaCl 11.7% (w/v) (AM9760G, Ambion) in PBS; sodium acrylate stock solution 33% (w/v) was freshly prepared by dissolving 1.9 g sodium acrylate in 5 ml of ddH$_2$O, 0.2 µm-filtered. For gelation, a gelling solution was prepared from 100 µl monomer solution, 1 µl 10% APS, and 1 µl 10% TEMED, vortexed, and used instantly. 50 µl of gelling solution was placed on a piece of parafilm in a humid chamber; the coverslips were placed on top, cells facing downwards, and incubated at 37 °C for 1 hr. After polymerisation, coverslips with the gel attached were transferred to a well of a 6sixwell plate with 1 ml of digestion solution 0.5% Triton X100, 1 mM EDTA pH 8 (AM9260G, Ambion), 50 mM Tris pH 8 (AM9855G, Ambion), and 1 M NaCl (AM9760G, Ambion) freshly supplemented with 10 µl proteinase K (P8107S, New England Biolabs). Plates were stored slightly tilted to ensure the coverslip was completely covered with digestion solution and incubated at room temperature overnight. Next day, gels were either already detached from the coverslip or could be easily detached with a spatula. Gels were transferred to a Petri dish and three rounds of expansion, each with 20 ml ddH$_2$O for 20 min, were done. After this, a picture was taken to measure the expansion factor. Gel mounting in the imaging chamber and imaging were done as described for UExM.

## Acknowledgements

The work was financed by a bilateral GACR/DFG grant (project IDs.: 21–19503 J and KR4017/9-1; to MZ and SK, respectively). We are grateful to the OMICS Proteomics BIOCEV core facility for its excellent technical service. We thank Mark Carrington (University of Cambridge, UK) for providing anti-TbMEX67. Manfred Alsheimer is acknowledged for providing advice and plasmids on HeLa cell transfections, Christian Janzen for the BB2 antibody, and Elisabeth Meyer-Natus and Tim Krüger for help with microscopy and image analysis (all University of Würzburg, Germany).

## Additional information

### Funding

| Funder | Grant reference number | Author |
| --- | --- | --- |
| Czech Science Foundation | 21-19503J | Martin Zoltner |
| Deutsche Forschungsgemeinschaft | KR4017/9-1 | Susanne Kramer |

| Funder | Grant reference number | Author |
|--------|------------------------|--------|

The funders had no role in study design, data collection and interpretation, or the decision to submit the work for publication.

### Author contributions
Johanna Odenwald, Bernardo Gabiatti, Conceptualization, Data curation, Formal analysis, Investigation, Writing – review and editing; Silke Braune, Siqi Shen, Data curation; Martin Zoltner, Conceptualization, Data curation, Supervision, Funding acquisition, Methodology, Writing – review and editing; Susanne Kramer, Conceptualization, Supervision, Funding acquisition, Writing - original draft, Project administration, Writing – review and editing

### Author ORCIDs
Bernardo Gabiatti ⬤ https://orcid.org/0000-0001-9690-2807
Martin Zoltner ⬤ http://orcid.org/0000-0002-0214-285X
Susanne Kramer ⬤ https://orcid.org/0000-0002-6302-2560

Reviewer #1 (Public review): https://doi.org/10.7554/eLife.95028.3.sa1
Reviewer #2 (Public review): https://doi.org/10.7554/eLife.95028.3.sa2
Author response https://doi.org/10.7554/eLife.95028.3.sa3

---

## Additional files

### Supplementary files
- MDAR checklist
- Supplementary file 1. List of all plasmids used in this work.
- Supplementary file 2. NUP TurboID proteomics data and statistical analysis.

### Data availability
All proteomics data have been deposited at the ProteomeXchange Consortium via the PRIDE partner repository with the dataset identifier PXD031245 (MEX67, NUP158, wt control, GFP control) and PXD047268 (NUP76, NUP96, NUP110).

The following datasets were generated:

| Author(s) | Year | Dataset title | Dataset URL | Database and Identifier |
|-----------|------|---------------|-------------|-------------------------|
| Zoltner M | 2023 | Probing the trypanosome nuclear pore by TurboID proximity labelling | https://proteomecentral.proteomexchange.org/cgi/GetDataset?ID=PXD047268 | ProteomeXchange, PXD047268 |
| Zoltner M | 2023 | Proximity labelling and cryomill affinity purification compared for *Trypanosoma brucei* proteins with unconfined, semi-confined and confined localisation | https://proteomecentral.proteomexchange.org/cgi/GetDataset?ID=PXD031245 | ProteomeXchange, PXD031245 |

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
