## [Editor Report · eLife assessment]

This **valuable** study demonstrates how proximity labeling with streptavidin can be used to boost fluorescence signals in otherwise hard-to-label regions of cells. The experimental verification of amplification of fluorescence near epitope tags in phase-separated compartments is **solid**, demonstrating enhanced signal-to-noise compared to immunofluorescence. This study will be of particular interest to those using correlative light and electron microscopy or expansion microscopy when the signal is limiting or inaccessible.

---

## [Referee Report · Reviewer #1 (Public review)]

I feel that the changes to the manuscript have significantly improved it. It's unfortunate that the single biotin/anti-biotin antibodies were not more illuminating but I think the attempts were worthwhile. My only comment is that the rebuttal to the second part of point 3 still does not fully deal with the issue. By marking proximal proteins other than the fusion with biotin, TurboID significantly increases the detectable signal, but it is formally no longer possible to be certain what the biotin is attached to. None of the controls that the authors suggest will actually give you certainty about what you are detecting while imaging. Mass spectrometry will give you an ensemble measurement of all the biotinylated proteins in the cell without being able to relate that back to what you are observing in a specific cellular region when you are imaging. Colocalization with a tagged protein/specific antibody could suggest that a portion of the signal could be attributable to the TurboID-biotin signal, but it could also be a tight binding partner or part of a larger protein complex. PLA assays would have similar issues- some of the protein could be labeled but it will be impossible to show what portion of the signal is attributable to the protein of interest and how much is attributable to other proximal proteins. I think the key thing here is that in this implementation, TurboID allows you to enhance the labeling of protein structures in cells, such as NUPS, but at the expense of certainty about the specific proteins you are labeling. I personally cannot think of a reasonable control that will allow you to avoid this issue. I feel that this point needs to be clearly made if people are going to use this method for signal enhancement, otherwise people may be misled about what they are actually looking at. The method should be useful, but the limitations need to be clear.

---

## [Referee Report · Reviewer #2 (Public review)]

I found the original paper to be of high quality and value. The revisions the authors have made (particularly with respect to the more cautious phraseology concerning the ability to track labelled proteins) are valuable additions. The other responses are well-argued and satisfactory to this reviewer.

---

## [Author Response]

The following is the authors’ response to the original reviews.

**Public Reviews:**

**Reviewer #1 (Public Review):**
Summary:In this work, Odenwald and colleagues show that mutant biotin ligases used to perform proximity-dependent biotin identification (TurboID) can be used to amplify signal in fluorescence microscopy and to label phase-separated compartments that are refractory to many immunofluorescence approaches. Using the parasite *Trypanosoma brucei*, they show that fluorescent methods such as expansion microscopy and CLEM, which require bright signals for optimal detection, benefit from the elevated signal provided by TurboID fusion proteins when coupled with labeled streptavidin. Moreover, they show that phase-separated compartments, where many antibody epitopes are occluded due to limited diffusion and potential sequestration, are labeled reliably with biotin deposited by a TurboID fusion protein that localizes within the compartment. They show successful labeling of the nucleolus, likely phase-separated portions of the nuclear pore, and stress granules. Lastly, they use a panel of nuclear pore-TurboID fusion proteins to map the regions of the *T. brucei* nuclear pore that appear to be phase-separated by comparing antibody labeling of the protein, which is susceptible to blocking, to the degree of biotin deposition detected by streptavidin, which is not.Strengths:Overall, this study shows that TurboID labelling and fluorescent streptavidin can be used to boost signal compared to conventional immunofluorescence in a manner similar to tyramide amplification, but without having to use antibodies. TurboID could prove to be a viable general strategy for labeling phase-separated structures in cells, and perhaps as a means of identifying these structures, which could also be useful.Weaknesses:However, I think that this work would benefit from additional controls to address if the improved detection that is being observed is due to the increased affinity and smaller size of streptavidin/biotin compared to IgGs, or if it has to do with the increased amount of binding epitope (biotin) being deposited compared to the number of available antibody epitopes. I also think that using the biotinylation signal produced by the TurboID fusion to track the location of the fusion protein and/or binding partners in cells comes with significant caveats that are not well addressed here, mostly due to the inability to discern which proteins are contributing to the observed biotin signal.To dissect the contributions of the TurboID fusion to elevating signal, anti-biotin antibodies could be used to determine if the abundance of the biotin being deposited by the TurboID is what is increasing detection, or if streptavidin is essential for this.

We agree with the reviewer, that it would be very interesting to distinguish whether the increase in signal comes from the multiple biotinylation sites or from streptavidin being a very good binder, or perhaps from both. However, this question is very hard to answer, as antibodies differ massively in their affinity to the antigen which is further dependent on the respective IF-conditions, and are therefore not directly comparible. Even if anti-biotin gives a better signal then anti-HA, this can be either caused by the increase in antigen-number (more biotin than HA-tag) or by the higher binding affinity, or by a combination of both, thus hard to distinguish. Nevertheless, we have tested monoclonal mouse anti-biotin targeting the (non-phase-separated) NUP158. We found the signal from the biotin-antibody to be much weaker than from anti-HA, indicating that, at least this particular biotin antibody, is not a very good binder in IF.

Alternatively, HaloTag or CLIP tagging could be used to see if diffusion of a small molecule tag other than biotin can overcome the labeling issue in phase-separated compartments. There are Halo-biotin substrates available that would allow the conjugation of 1 biotin per fusion protein, which would allow the authors to dissect the relative contributions of the high affinity of streptavidin from the increased amount of biotin that the TurboID introduces.

This is a very good idea, as in this case, the signals are both from streptavidin and are directly comparable. We expressed NUP158 with HaloTag and added PEG-biotin as a Halo ligand. However, PEG-biotin is poorly cell-permeable, and is in general only used on lysates. In trypanosomes, cell permeability is particular restricted, and even Halo-ligands that are considered highly cell-penetrant give only a weak signal. Even after over-night incubation, we could not get any signal with PEG-biotin. Our control, the TMR-ligand 647, gave a weak nuclear pore staining, confirming the correct expression and function of the HaloTag-NUP158.

The idea of using the biotin signal from the TurboID fusion as a means to track the changing localization of the fusion protein or the location of interacting partners is an attractive idea, but the lack of certainty about what proteins are carrying the biotin signal makes it very difficult to make clear statements. For example, in the case of TurboID-PABP2, the appearance of a biotin signal at the cell posterior is proposed to be ALPH1, part of the mRNA decapping complex. However, because we are tracking biotin localization and biotin is being deposited on a variety of proteins, it is not formally possible to say that the posterior signal is ALPH1 or any other part of the decapping complex. For example, the posterior labeling could represent a localization of PABP2 that is not seen without the additional signal intensity provided by the TurboID fusion. There are also many cytoskeletal components present at the cell posterior that could be being biotinylated, not just the decapping complex. Similar arguments can be made for the localization data pertaining to MLP2 and NUP65/75. I would argue that the TurboID labeling allows you to enhance signal on structures, such as the NUPs, and effectively label compartments, but you lack the capacity to know precisely which proteins are being labeled.

We fully agree with the reviewer, that tracking proteins by streptavidin imaging alone is problematic, because it cannot distinguish, which protein is biotinylated. We therefore used words like “likely” in the description of the data. However, we still think, it is a valid method, as long as it is confirmed by an orthogonal method. We have added this paragraph to the end of this chapter:

“Importantly, tracking of proteins by streptavidin imaging requires orthogonal controls, as the imaging alone does not provide information about the nature of the biotinylated proteins. These can be proximity ligation assay, mass spectrometry or specific tagging visualisation of protein suspects by fluorescent tags. Once these orthogonal controls are established for a specific tracking, streptavidin imaging is an easy and cheap and highly versatile method to monitor protein interactions in a specific setting.”

**Reviewer #2 (Public Review):**
Summary:The authors noticed that there was an enhanced ability to detect nuclear pore proteins in trypanosomes using a streptavidin-biotin-based detection approach in comparison to conventional antibody-based detection, and this seemed particularly acute for phase-separated proteins. They explored this in detail for both standard imaging but also expansion microscopy and CLEM, testing resolution, signal strength, and sensitivity. An additional innovative approach exploits the proximity element of biotin labelling to identify where interacting proteins have been as well as where they are.Strengths:The data is high quality and convincing and will have obvious application, not just in the trypanosome field but also more broadly where proteins are tricky to detect or inaccessible due to phase separation (or some other steric limitations). It will be of wide utility and value in many cell biological studies and is timely due to the focus of interest on phase separation, CLEM, and expansion microscopy.

Thank you! We are glad you liked it.

**Reviewer #3 (Public Review):**
Summary:The authors aimed to investigate the effectiveness of streptavidin imaging as an alternative to traditional antibody labeling for visualizing proteins within cellular contexts. They sought to address challenges associated with antibody accessibility and inconsistent localization by comparing the performance of streptavidin imaging with a TurboID-HA tandem tag across various protein localization scenarios, including phase-separated regions. They aimed to assess the reliability, signal enhancement, and potential advantages of streptavidin imaging over antibody labeling techniques.Overall, the study provides a convincing argument for the utility of streptavidin imaging in cellular protein visualization. By demonstrating the effectiveness of streptavidin imaging as an alternative to antibody labeling, the study offers a promising solution to issues of accessibility and localization variability. Furthermore, while streptavidin imaging shows significant advantages in signal enhancement and preservation of protein interactions, the authors must consider potential limitations and variations in its application. Factors such as the fact that tagging may sometimes impact protein function, background noise, non-specific binding, and the potential for off-target effects may impact the reliability and interpretation of results. Thus, careful validation and optimization of streptavidin imaging protocols are crucial to ensure reproducibility and accuracy across different experimental setups.Strengths:- Streptavidin imaging utilizes multiple biotinylation sites on both the target protein and adjacent proteins, resulting in a substantial signal boost. This enhancement is particularly beneficial for several applications with diluted antigens, such as expansion microscopy or correlative light and electron microscopy.- This biotinylation process enables the identification and characterization of interacting proteins, allowing for a comprehensive understanding of protein-protein interactions within cellular contexts.Weaknesses:- One of the key advantages of antibodies is that they label native, endogenous proteins, i.e. without introducing any genetic modifications or exogenously expressed proteins. This is a major difference from the approach in this manuscript, and it is surprising that this limitation is not really mentioned, let alone expanded upon, anywhere in the manuscript. Tagging proteins often impacts their function (if not their localization), and this is also not discussed.- Given that BioID proximity labeling encompasses not only the protein of interest but also its entire interacting partner history, ensuring accurate localization of the protein of interest poses a challenge.- The title of the publication suggests that this imaging technique is widely applicable. However, the authors did not show the ability to track the localization of several distinct proteins on the same sample, which could be an additional factor demonstrating the outperformance of streptavidin imaging compared with antibody labeling. Similarly, the work focuses only on small 2D samples. It would have been interesting to be able to compare this with 3D samples (e.g. cells encapsulated in an extracellular matrix) or to tissues.
**Recommendations for the authors:**
To enhance the assessment from 'incomplete' to 'solid', the reviewers recommend that the following major issues be addressed:Major issues:(1) Anti-biotin antibodies in combination with TurboID labeling should be used to compare the signal/labelling penetrance to streptavidin results. That would show if elevated biotin deposition matters, or if it is really the smaller size, more fluors, and higher affinity of streptavidin that's making the difference.

We agree with the reviewer, that it would be very interesting to distinguish whether the increase in signal comes from the multiple biotinylation sites or from streptavidin being a very good binder, or perhaps from both, and whether the size matters (IgG versus streptavidin). However, this question is very hard to answer, as antibodies differ massively in their affinity to the antigen. Thus, even if antibiotin would give a better signal then anti-HA, this could be either caused by the increase in antigen-number (more biotin than HA-tag) or by the better binding affinity, or by a combination, and it would not allow to truly answer the question. We have now tested anti-biotin antibodies, also in repsonse to reviewer 1, and got a much poorer signal in comparison to anti-HA or streptavidin.

Please note that we made another attempt using nanobodies to target phase-separated proteins, to see, whether size matters (Fig. 2I). The nanobody did not stain Mex67 at the nuclear pores, but gave a weak nucelolar signal for NOG1, which may suggest that the nanobody can slightly better penetrate than IgG, but it does not rule out that the nanobody simply binds with higher affinity. Reviewer 1 has suggested to use the Halo Tag with PEG-biotin: this would indeed allow to directly compare the streptavidin signal caused by the TurboID with a single biotin added by the Halo tag. Unfortunately, the PEG-biotin does not penetrate trypanosome cells. In conclusion, we are not aware of a method that would allow to establish why streptavidin but not IgGs can penetrate to phase separated areas. We therefore prefer to not overinterpret our data, but stick to what is supported by the data: “the inability to label phase-separated areas is not restricted to anti-HA but applies to other antibodies”.

(3) Figure 4 A-B. The validity of claiming the correct localization demonstrated by streptavidin imaging comes into question, especially when endogenous fluorescence, via the fusion protein, remains undetectable (as indicated by the yellow arrow at apex).

In this figure, the streptavidin imaging does NOT show the correct localisation of the bait protein, but it does show proteins from historic interactions that have a distinct localisation to the bait. We had therefore introduced this chapter with the paragraph below, to make sure, the reader is aware of the limitations (which we also see as an opportunity, if properly controlled):

“We found that in most cases, streptavidin labelling faithfully reflects the steady state localisation of a bait protein, e.g., the localisation resembles those observed with immunofluorescence or direct fluorescence imaging of GFP-fusion proteins. For certain bait proteins, this is not the case, for example, if the bait protein or its interactors have a dynamic localisation to distinct compartments, or if interactions are highly transient. It is thus essential to control streptavidin-based de novo localisation data by either antibody labelling (if possible) or by direct fluorescence of fusion-proteins for each new bait protein.”

In particular, on lines 450-460, there's a fundamental issue with the argument put forward here. It is not possible to formally know that the posterior labeling is ALPH1 vs. another part of the decapping complex that was associated with PABP2-Turbo, or if the higher detection capacity of the Turbo-biotin label is uncovering a novel localization of the PABP2. While it is likely that it is ALPH1, it is not possible to rule out other possibilities with this approach. These issues should be discussed here and more generally the possibility of off-target labeling with this approach should be addressed in the discussion.

We fully agree with the reviewer, that tracking proteins by streptavidin imaging alone is problematic, because it cannot distinguish, which protein is biotinylated. We therefore used words like “likely” in the description of the data. However, we still think, it is a valid method, as long as it is back-uped by an orthogonal method. We have added this paragraph to the end of this chapter:

“Importantly, tracking of proteins by streptavidin imaging requires orthogonal controls, as the imaging alone does not provide information about the nature of the biotinylated proteins. These can be proximity ligation assay, mass spectrometry or specific tagging visualisation of protein suspects by fluorescent tags. Once these orthogonal controls are established for a specific tracking, streptavidin imaging is an easy and cheap and highly versatile method to monitor protein interactions in a specific setting.”

(4) More discussion and acknowledgment of the general limitations in using tagged proteins are needed to balance the manuscript, especially if the hope is to draw a comparison with antibody labeling, which works on endogenous proteins (not requiring a tag). For example: (a) tagging proteins requires genetic/molecular work ahead of time to engineer the constructs and/or cells if trying to tag endogenous proteins; (b) tagged proteins should technically be validated in rescue experiments to confirm the tag doesn't disrupt function in the cell/tissue/context of interest; and (c) exogenous tagged proteins compete with endogenous untagged proteins, which can complicate the interpretation of data.

We have added this paragraph to the first paragraph of the discussion part:

“Like many methods that are frequently used in cell- and molecular biology, streptavidin imaging is based on the expression of a genetically engineered fusion protein: it is essential to validate both, function and localisation of the TurboID-HA tagged protein by orthogonal methods. If the fusion protein is non-functional or mis-localised, tagging at the other end may help, but if not, this protein cannot be imaged by streptavidin imaging. Likewise, target organisms not amenable to genetic manipulation, or those with restricted genetic tools, are not or less suitable for this method.”

Also, we like to point out that for non-mainstream organisms like trypanosomes, antibodies are not commercially available and often genetic manipulation is more time-efficient and cheaper than the production of antiserum against the target protein.

Also, the introduction would ideally be more general in scope and introduce the pros and cons of antibody labeling vs biotin/streptavidin, which are mentioned briefly in the discussion. The fact that the biotin-streptavidin interaction is ~100-fold higher affinity than an IgG binding to its epitope is likely playing a key role in the results here. The difference in size between IgG and streptavidin, the likelihood that the tetrameric streptavidin carries more fluors than a IgG secondary, and the fact that biotin can likely diffuse into phase-separated environments should be clearly stated. The current introduction segues from a previous paper that a more general audience may not be familiar with.

We have now included this paragraph to the introduction:

“It remains unclear, why streptavidin was able to stain biotinylated proteins within these antibody inaccessible regions, but possible reasons are: (i) tetrameric streptavidin is smaller and more compact than IgGs (60 kDa versus a tandem of two IgGs, each with 150 kDa) (ii) the interaction between streptavidin and biotin is ~100 fold stronger than a typical interaction between antibody and antigen and (iii) streptavidin contains four fluorophores, in contrast to only one per secondary IgG.”

Minor issues:The copy numbers of the HA and Ty1 epitope tags vary depending on the construct being used. For example, Ty1 is found as a single copy tag in the TurboID tag, but on the mNeonGreen tag there are 6 copies of the epitope. It makes it hard to know if differences in detection are due to variations in copies of the epitope tags. Line 372-374: can the authors explain why they chose to use nanobodies in this case? It would be great to show the innate mNeonGreen signal in 2K to compare to the Ty1 labeling. The presence of 6 copies of the Ty1 epitope could be essential to the labeling seen here.

We agree with the reviewer, that these data are a bit confusing. We have now removed Figure 3K, as it is the only construct with 6 Ty1 instead of one, and it does not add to the conclusions. (the mNeonsignal is entirely in the nucleolus, as shown by Tryptag). We have also added an explanation why we used nanobodies (“The absence of a nanobody signal rules out that its simply the size of IgGs that prevents the staining of Mex67 at the nuclear pores, as nanobodies are smaller than (tetrameric) streptavidin”). However, as stated above, we prefer not to overinterpret the data, as signals from different antibodies/nanobodies – antigen combinations are not comparable. Important to us was to stress that the absence of signal in phase-separated areas is NOT restricted to the anti-HA antibody, which is clearly supported by the data.

What is the innate streptavidin background labeling look like in cells that are not carrying a TurboID fusion, from the native proteins that are biotinylated? That should be discussed.

We have now included the controls without the TurboID fusions for trypanosomes and HeLa cells: “Wild type cells of both Trypanosomes and human showed only a very low streptavidin signal, indicating that the signal from naturally biotinylated proteins is neglectable (Figure S8 in supplementary material).”

Line 328-331: This is likely to be dependent on whether or not the protein moves to different localizations within the cell.

True, we agree, and we have added this paragraph:

“The one exception are very motile proteins that produce a “biotinylation trail” distinct to the steady state localisation; these exceptions, and how they can be exploited to understand protein interactions, are discussed in chapter 4 below. “

Line 304-305: Does biotin supplementation not matter at all?

No, we never saw any increase in biotinylation when we added extra biotin to trypanosomes. The 0.8 µM biotin concentration in the medium were sufficient.

Line 326-327: Was the addition of biotin checked for enhancement in the case of the mammalian NUP98? I would argue that there is a significant number of puncta in Figure 1D that are either green or magenta, not both. The amount of extranuclear puncta in the HA channel is also difficult to explain. Biotin supplementation to 500 µM was used in mammalian TurboID experiments in the original Nature Biotech paper- perhaps nanomolar levels are too low.

We now tested HeLa cells with 500 µM Biotin and saw an increase in signal, but also in background; due to the increased background we conclude that low biotin concentrations are more suitable . We have also repeated the experiment using 4HA tags instead of 1HA, and we found a minor improvement in the antibody signal for NUP88 (while the phase separated NUP54 was still not detectable). We have replaced the images in Figure 1D (NUP88) and also in Figure 2F (NUP54) with improved images and using 4HA tags. However, we like to note that single nuclear pore resolution is beyond what can be expected of light microscopy.

Line 371: In 2I, I see a signal that looks like the nucleus, similar to the Ty1 labeling in 2G, so I don't think it's accurate to say that that Mex67 was "undetectable". Does the serum work for blotting?

Thank you, yes, “undetectable” was not the correct phrase here. Mex67 localises to the nuclear pores, to the nuceoplasm and to the nucleolus (GFP-tagging or streptavidin). Antibodies, either to the tag or to the endogenous proteins, fail to detect Mex67 at the nuclear pores and also don’t show any particular enrichment in the nucleolus. They do, however, detect Mex67 in the (not-phase-separated) area of the nucleoplasm. We have changed the text to make this clearer. The Mex67 antiserum works well on a western blot (see for example: Pozzi, B., Naguleswaran, A., Florini, F., Rezaei, Z. & Roditi, I. The RNA export factor TbMex67 connects transcription and RNA export in *Trypanosoma brucei* and sets boundaries for RNA polymerase I. *Nucleic Acids Res.*
**51**, 5177–5192 (2023))

Line 477: "lacked" should be "lagged".

Thank you, corrected.

Line 468-481: My previous argument holds here - how do you know that the difference in detection here is just a matter of much higher affinity/quantity of binding partner for the avidin?

See answer to the second point of (3), above.

483-491: Same issue - without certainty about what the biotin is on, this argument is difficult to make.

See answer to the second point of (3), above.

Line 530: "bone-fine" should be "bonafide"

Thank you, corrected.

Line 602: biotin/streptavidin labeling has been used for expansion microscopy previously (Sun, Nature Biotech 2021; PMID: 33288959).

Thank you, we had overlooked this! We have now included this reference and describe the differences to our approach clearer in the discussion part:

“Fluorescent streptavidin has been previously used in expansion microscopy to detect biotin residues in target proteins produced by click chemistry (Sun et al., 2021). However, to the best of our knowledge, this is the first report that employs fluorescent streptavidin as a signal enhancer in expansion microscopy and CLEM, by combining it with multiple biotinylation sites added by a biotin ligase. Importantly, for both CLEM and expansion, streptavidin imaging is the only alternative approach to immunofluorescence, as denaturing conditions associated with these methods rule out direct imaging of fluorescent tags.”